# The CCR4–NOT complex maintains liver homeostasis through mRNA deadenylation

Akinori Takahashi[1,*], Toru Suzuki[5,*], Shou Soeda[1,*], Shohei Takaoka[1], Shungo Kobori[2], Tomokazu Yamaguchi[3], Haytham Mohamed Aly Mohamed[1], Akiko Yanagiya[1], Takaya Abe[4], Mayo Shigeta[4], Yasuhide Furuta[4], Keiji Kuba[3], Tadashi Yamamoto[1,5]

**The biological significance of deadenylation in global gene expression is not fully understood. Here, we show that the CCR4–NOT deadenylase complex maintains expression of mRNAs, such as those encoding transcription factors, cell cycle regulators, DNA damage response–related proteins, and metabolic enzymes, at appropriate levels in the liver. Liver-specific disruption of *Cnot1*, encoding a scaffold subunit of the CCR4–NOT complex, leads to increased levels of mRNAs for transcription factors, cell cycle regulators, and DNA damage response–related proteins because of reduced deadenylation and stabilization of these mRNAs. CNOT1 suppression also results in an increase of immature, unspliced mRNAs (pre-mRNAs) for apoptosis-related and inflammation-related genes and promotes RNA polymerase II loading on their promoter regions. In contrast, mRNAs encoding metabolic enzymes become less abundant, concomitant with decreased levels of these pre-mRNAs. Lethal hepatitis develops concomitantly with abnormal mRNA expression. Mechanistically, the CCR4–NOT complex targets and destabilizes mRNAs mainly through its association with Argonaute 2 (AGO2) and butyrate response factor 1 (BRF1) in the liver. Therefore, the CCR4–NOT complex contributes to liver homeostasis by modulating the liver transcriptome through mRNA deadenylation.**

## Introduction

In mammals, the liver is essential to control energy intake and expenditure so as to maintain organismal energy homeostasis. Disruption of liver function results in metabolic disorders and diseases, including hepatitis, hepatic cirrhosis, liver cancer, multiple organ failure, and death (Malhi et al, 2010). Precise regulation of gene expression is required for liver homeostasis. In the liver,

mRNAs encoding metabolic enzymes are regulated by transcription factors (TFs) such as hepatocyte nuclear factors, peroxisome proliferator–activated receptor α, and sterol regulatory element–binding transcription factor 1 (Horton et al, 2002; Desvergne et al, 2006; Martinez-Jimenes et al, 2010). Although it is widely accepted that transcription contributes to gene expression control, the importance of posttranscriptional mechanisms, including mRNA decay, for appropriate gene expression is increasingly appreciated (Garneau et al, 2007; Schoenberg et al, 2012). For instance, DICER, an enzyme for processing microRNAs, suppresses hepatocyte growth and fetal stage–specific genes (Sekine et al, 2009). *miR-122*, the most abundant microRNA in the liver, suppresses tumor-progressive genes to prevent hepatocarcinogenesis (Hsu et al, 2012; Tsai et al, 2012). Thus, both transcriptional and posttranscriptional mechanisms participate in liver homeostasis.

Shortening of polyadenosine (poly(A)) tails by deadenylation is the initial step in the degradation of most mRNAs (Garneau et al, 2007; Schoenberg et al, 2012; Mugridge et al, 2018). Removal of the poly(A) tail facilitates decapping of the 5' cap structure, leading to 5'-3' exonuclease-mediated mRNA decay. After removal of the poly(A) tail, mRNA degradation is also carried out from the 3'-5' end by the exosome complex, containing 3'-5' exonucleases. The CCR4–NOT (carbon catabolite repression 4–negative on TATA less) complex, the major deadenylase in mammals, shortens mRNA poly(A) tails (Temme et al, 2010; Nousch et al, 2013; Shirai et al, 2014). This complex comprises at least eight subunits, CNOT1-3, either CNOT6 or CNOT6L, either CNOT7 or CNOT8, and CNOT9-11 (Collart & Timmers, 2004; Shirai et al, 2014). CNOT6/6L (CCR4a/b) and CNOT7/8 (CAF1a/b) catalytic subunits belong to the exonuclease–endonuclease–phosphatase family and the DEDD (Asp-Glu-Asp-Asp) family, respectively (Goldstrohm & Wickens, 2008). CNOT1 acts as a scaffold for the other subunits (Bai et al, 1999; Maillet et al, 2000; Collart & Timmers, 2004; Winkler & Whale, 2013). CNOT1 has several domains, including a tristetraprolin (TTP)–binding domain, the middle domain of eukaryotic initiation factor 4G (MIF4G)

[1]Cell Signal Unit, Okinawa Institute of Science and Technology Graduate University, Okinawa, Japan   [2]Nucleic Acid Chemistry and Engineering Unit, Okinawa Institute of Science and Technology Graduate University, Okinawa, Japan   [3]Department of Biochemistry and Metabolic Science, Graduate School of Medicine, Akita University, Akita, Japan   [4]Laboratory for Animal Resources and Genetic Engineering, RIKEN Center for Biosystems Dynamics Research, Kobe, Japan   [5]Laboratory for Immunogenetics, Center for Integrative Medical Sciences, RIKEN, Yokohama City, Kanagawa, Japan

Correspondence: tadashi.yamamoto@oist.jp; toru.suzuki.ff@riken.jp
*Akinori Takahashi, Toru Suzuki, and Shou Soeda contributed equally to this work

domain, which interacts with CNOT6/6L/7/8 enzymatic subunits, a DUF3819 domain that interacts with CNOT9, and a NOT1 domain that is required for interaction with CNOT2/3 ((Basquin et al., 2012; Petit et al., 2012); (Bawankar et al., 2013); (Boland et al., 2013; Chen et al., 2014; Fabian et al., 2013; Ukleja et al., 2016)). CNOT1 functions as the scaffold for RNA-binding proteins that recruit the complex to the 3′-UTR of target mRNAs so as to degrade them. Those RNA-binding proteins include the miRNA-induced silencing complex (miRISC), which contains AGO family proteins and GW182 as core proteins, the TTP family of AU-rich element (ARE)–binding proteins (TTP and BRF1/2), ROQUIN, cytoplasmic polyadenylation element-binding proteins, and NANOS (Fabian et al, 2011; Hosoda et al, 2011; Adachi et al, 2014; Bhandari et al, 2014; Ogami et al, 2014; Sgromo et al, 2017).

The CCR4–NOT complex participates in the regulation of cell viability, energy metabolism, and tissue development via mRNA degradation in a cell-type– and tissue-specific manner (Shirai et al, 2014). Suppression of the complex results in various forms of cell death. Increased p53 or Caspase-4 leads to apoptosis in *Cnot3*-depleted murine B-cells or *Cnot2*-depleted HeLa cells, respectively (Ito et al, 2011a; Inoue et al, 2015). RIPK3-mediated necroptosis is markedly induced in *Cnot3*-depleted MEFs (Suzuki et al, 2015), whereas autophagy-independent ATG7/Trp53–mediated cell death is observed in heart-specific, *Cnot1/3*-deficient mice (Yamaguchi et al, 2018). In metabolic tissues, the CCR4–NOT complex deadenylates mRNAs encoding metabolic proteins such as PDK4 and UCP1 (Morita et al, 2011; Takahashi et al, 2015). Excess nutrients affect CCR4–NOT complex activity and expression of mRNAs that encode metabolic proteins relevant to obesity (Morita et al, 2011; Takahashi et al, 2015). The CCR4–NOT complex also regulates adipocyte function and liver functional maturation (Li et al, 2017; Suzuki et al, 2019; Takahashi et al, 2019). However, the mechanisms by which deadenylation regulates tissue-specific mRNA profiles and homeostasis have not been fully addressed.

Our previous study showed that CCR4–NOT complex–mediated decay of immature liver mRNAs is required for liver functional maturation (Suzuki et al, 2019). In this study, we address the effects of mRNA deadenylation on global gene expression and tissue homeostasis in adult mature liver. We find that the CCR4–NOT complex maintains the liver transcriptome via its deadenylase activity. In mature liver, the CCR4–NOT complex binds to mRNAs encoding TFs, cell cycle regulators, DNA damage response-related proteins, and liver function–related proteins. Consequently, disruption of the CCR4–NOT function through CNOT1 depletion induces aberrant gene expression that is associated with lethal hepatitis. Therefore, the ability of the CCR4–NOT complex to maintain the liver transcriptome is crucial for liver homeostasis.

# Results

### Liver-specific disruption of *Cnot1* causes lethal hepatitis associated with elongated mRNA poly(A) tails

Suppression of CNOT1 largely abrogated deadenylase activity (Temme et al, 2010; Ito et al, 2011; Nousch et al, 2013; Mostafa et al,

2020), suggesting that CNOT1 is an essential scaffold subunit in the CCR4–NOT complex in vivo. We generated conditional KO mice for *Cnot1* (*Cnot1fl/fl* mice) by inserting loxP sequences into the *Cnot1* gene locus so that exons 20 and 21 were deleted (Fig S1A and B). Exons 20 and 21 encode amino acids 711-826 in CNOT1 protein and are located N-terminal to the TTP-binding domain (Fig S1A and B). Successful insertion of loxP sequences and generation of the KO allele after Cre-mediated recombination were confirmed by PCR analysis (Fig S1C and D). When we crossed heterozygous *Cnot1-KO* (*Cnot1+/−*) males and females, wild-type and *Cnot1+/−* mice were born at an ~1:1 ratio and grew normally to adulthood (Fig S1E). Homozygous *Cnot1-KO* (*Cnot1−/−*) mice were not obtained after embryonic day 8.5, indicating that *Cnot1−/−* mice die in embryo (Fig S1E). *Cnot1+/−* mice were not born at Mendelian frequencies, suggesting that *Cnot1* haploinsufficiency partly affects mouse embryonic development. The detailed reasons are to be addressed. To understand the impact of CCR4–NOT complex–dependent deadenylation on gene expression and homeostasis in the liver, we generated liver-specific *Cnot1*-KO mice (*Cnot1-LKO* mice). We obtained *Cnot1fl/fl;Alb-CreERT2* mice by crossing *Cnot1fl/+;Alb-CreERT2* pairs. To induce deletion of the *Cnot1* gene, *Cnot1fl/fl;Alb-CreERT2* mice were fed with a tamoxifen-containing diet. Tamoxifen-fed *Cnot1fl/fl;Alb-CreERT2* mice were used as *Cnot1-LKO* mice (see details in the Materials and Methods section).

CNOT1 protein levels decreased in the livers of *Cnot1-LKO* mice (Fig 1A). We first investigated whether *Cnot1* suppression affects liver function and causes physiological disorders. *Cnot1-LKO* mice had pale-colored livers and swollen gallbladders (Fig 1B) and died within 17 d after tamoxifen feeding (Fig 1C). 2 wk after tamoxifen feeding, body weight and circulating blood glucose were significantly lower in *Cnot1-LKO* mice than in control mice (*Cnot1fl/fl* or *Alb-CreERT2* mice) (Fig 1D and E), although liver mass was similar (Fig 1F). Histological diagnosis using hematoxylin and eosin (H&E)–stained liver sections revealed hepatocyte necrotic death and infiltration of immune cells in livers of *Cnot1-LKO*, but not control mice (Fig 1G and Table 1). Steatosis was hardly detected in livers of *Cnot1-LKO* mice (Table 1). Consistent with this, levels of inflammation-related and cell death–related mRNAs increased significantly in the livers of *Cnot1-LKO* mice (Fig 1H). We also detected an increase of phosphorylated-JNK, BAX, and Cleaved Caspase-3 using immunoblot analysis (Fig 1A). The appearance of Cleaved Caspase-3–positive cells in immunohistochemistry indicated apoptotic death of hepatocytes in livers from *Cnot1-LKO* mice (Fig 1I). Biochemical analysis of blood showed that alanine transaminase (ALT), aspartate aminotransferase (AST), alkaline phosphatase (ALP), and lactate dehydrogenase (LDH) levels were strongly elevated in *Cnot1-LKO* mice (Table 2). These data suggested that *Cnot1-LKO* mice died of severe hepatitis.

To examine the effects of CNOT1 suppression on the length of RNA poly(A) tails, we compared poly(A) tail lengths of bulk RNAs in the livers of control and *Cnot1-LKO* mice. In the livers of control mice (*Cnot1fl/fl* or *Alb-CreERT2*), poly(A) tail lengths of ~60 nucleotides (nt) were predominant (Fig 2A). In the livers from *Cnot1-LKO* mice, the population of mRNAs with poly(A) tails longer than 70 nt increased dramatically, whereas those with poly(A) tails of 30–70 nt decreased (Fig 2A). To verify accumulation of long poly(A) mRNAs in livers from *Cnot1-LKO* mice, we examined poly(A) tail lengths of

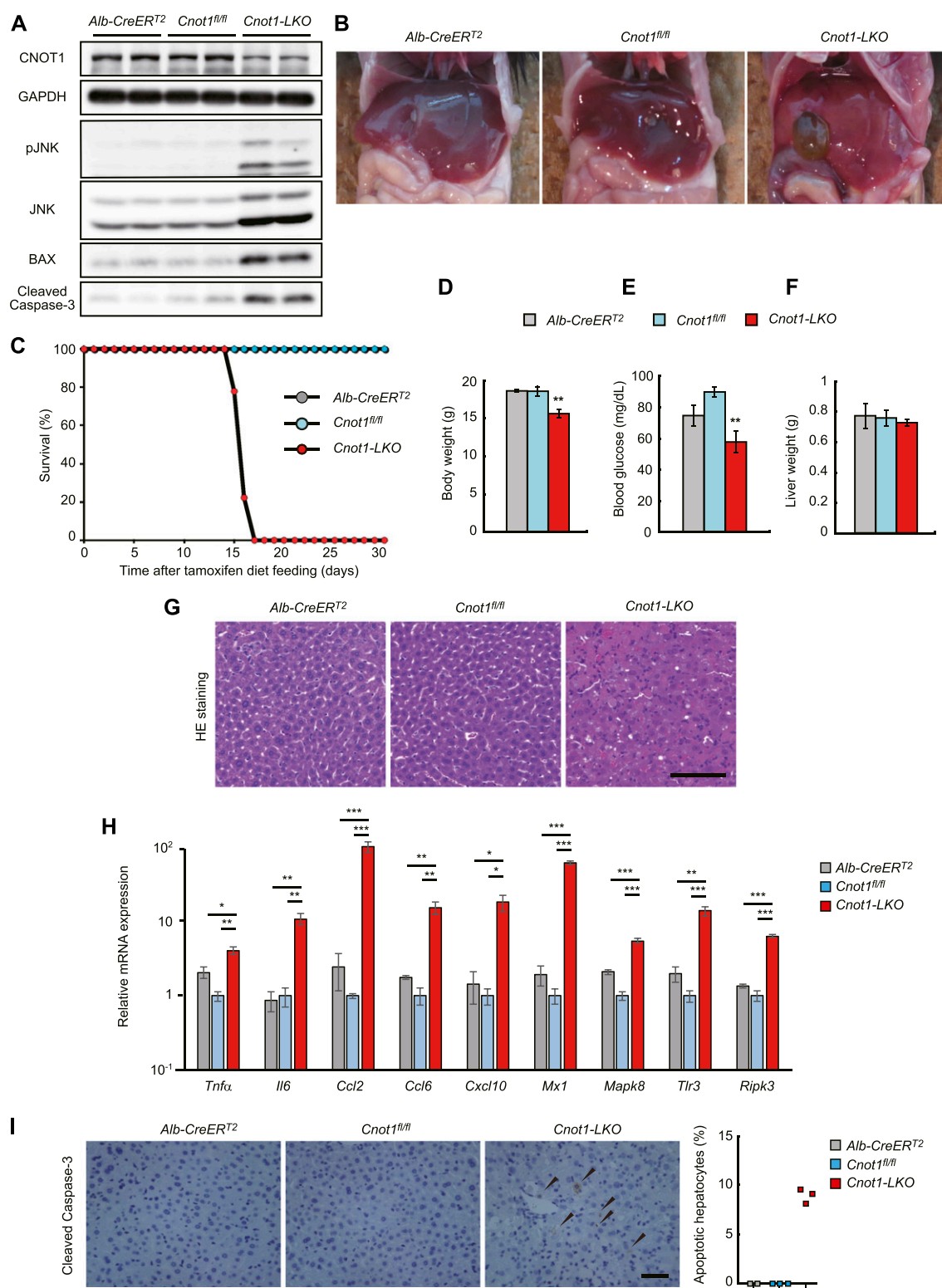

**Figure 1. Liver-specific disruption of *Cnot1* causes lethal hepatitis.**
**(A)** Immunoblotting of the indicated molecules in liver lysates from *Alb-CreER^T2^*, *Cnot1^fl/fl^*, and *Cnot1-LKO* mice. **(B)** Gross appearance of livers from *Alb-CreER^T2^*, *Cnot1^fl/fl^*, and *Cnot1-LKO* mice.
**(C)** Survival curves of *Alb-CreER^T2^* (n = 4), *Cnot1^fl/fl^* (n = 9), and *Cnot1-LKO* (n = 9) mice after a tamoxifen-containing diet. **(D, E, F)** Body weights (D), circulating blood glucose levels (E), and liver weights (F) of *Alb-CreER^T2^* (n = 4), *Cnot1^fl/fl^* (n = 7), and *Cnot1-LKO* (n = 9) mice. **(G)** H&E staining of livers from *Alb-CreER^T2^*, *Cnot1^fl/fl^*, and *Cnot1-LKO* mice. Scale bar, 100 μm. **(H)** Quantitative real-time PCR (qRT-PCR) analysis of the indicated mRNAs in livers from *Alb-CreER^T2^*, *Cnot1^fl/fl^*, and *Cnot1-LKO* mice (n = 4). The *Rplp0* mRNA level was used for normalization. **(I)** Immunohistochemistry for Cleaved Caspase-3 in livers from *Alb-CreER^T2^*, *Cnot1^fl/fl^*, and *Cnot1-LKO* mice. Scale bar, 100 μm. Right graph shows percentages of Cleaved Caspase-3–positive hepatocytes. Three different fields (total of ~500 cells) in each section were counted (*Alb-CreER^T2^*; n = 2, *Cnot1^fl/fl^* and *Cnot1-LKO* mice; n = 3). Values in graphs represent means ± SEM. Unpaired $t$ test, *$P < 0.05$, **$P < 0.01$, ***$P < 0.001$.

**Table 1.  Histopathological analysis of livers.**

| Sample name | Inflammation[a] | Steatosis[b] | Hepatocellular necrosis[c] | Histological score (0–9) |
|---|---|---|---|---|
| Cnot1-LKO 1 | 3 | 0 | 3 | 6 |
| Cnot1-LKO 2 | 3 | 0 | 3 | 6 |
| Cnot1 (fl/fl) 1 | 0 | 0 | 0 | 0 |
| Cnot1 (fl/fl) 2 | 0 | 0 | 0 | 0 |
| Alb-CreERT2 1 | 0 | 1 | 0 | 1 |
| Alb-CreERT2 2 | 0 | 0 | 0 | 0 |

[a]0, no inflammation; 1, mild lymphocytic infiltration in the portal triad; 2, severe lymphocytic infiltration in portal triad; 3, extended infiltration of lymphocytes throughout the liver.
[b]0, no steatosis; 1, microsteatosis; 2, microsteatosis and mild macrosteatosis; 3, severe macrosteatosis.
[c]0, no necrosis; 1, mild necrosis; 2, moderate necrosis; 3, severe necrosis.

individual mRNAs (*Cox4i1*, *Gapdh*, *Pdk4*, and *Ttr* mRNAs). In the livers from *Cnot1-LKO* mice, these mRNAs all had longer poly(A) tails than those from control mice (Fig 2B). These data suggest that CCR4–NOT complex–mediated mRNA deadenylation is critical for liver function and homeostasis.

### Increased levels of TF-, cell cycle- and DNA damage-mRNAs and decreased levels of liver function–related mRNAs in the livers of *Cnot1-LKO* mice

We compared mRNA expression profiles in the livers of control (*Cnot1*$^{fl/fl}$) and *Cnot1-LKO* mice by performing total RNA sequencing (RNA-seq). Using RNA-seq data, Fragments Per Kilobase of exons per Million mapped sequence reads (FPKMs) of genes (*Gene* FPKMs, where "Gene" designates any gene of interest) were calculated. The results showed that 8,116 mRNAs increased and 703 decreased more than twofold in the livers of *Cnot1-LKO* mice, compared with those of controls (Fig 3A and Table S1). Gene ontology (GO) analysis showed that GO terms "transcription," "cell cycle," and "cellular response to DNA damage stimulus" were enriched among increased mRNAs, whereas GO terms "oxidation–reduction process" and "lipid metabolic process" were enriched among decreased mRNAs (Fig 3B and Table S2). Consistent with hepatic cell death and inflammation (Fig 1), "apoptosis" and "immune system process" were also significantly enriched GO terms (Table S2). Furthermore, FPKM distributions of mRNAs belonging to "transcription," "cell cycle," or "cellular response to DNA damage stimulus"; TF-, cell cycle-, or DNA damage-mRNAs were significantly higher in the livers from *Cnot1-LKO* mice, compared with those of controls (*Cnot1*$^{fl/fl}$ or *Alb-CreER*$^{T2}$) (Fig S2A). In contrast, there was no significant difference in FPKM distribution of mRNAs belonging to GO terms

"oxidation–reduction process" and "lipid metabolic process" in the livers from control and *Cnot1-LKO* mice, indicating that only some of the mRNAs that encode molecules involved in "oxidation–reduction process" and "lipid metabolic process" decreased in *Cnot1-LKO* mice (Fig S2A). We verified these expression differences using quantitative real-time PCR (qRT-PCR) analysis and showed that TF-, cell cycle-, and DNA damage-mRNAs increased in the livers from *Cnot1-LKO* mice (Fig 3C). In contrast, mRNAs encoding molecules for oxidation–reduction process, lipid metabolic process, and other liver-related functions decreased in the livers from *Cnot1-KO* livers (Fig 3C). We performed poly(A) tail analyses on several TF-, DNA damage-, and cell cycle-mRNAs. The results showed that mRNAs encoding TFs (*Trp53* and *Jun*), DNA damage response–related molecules (*Bbc3* and *Brca1*), and cell cycle regulators (*Cdt1* and *Cdc25a*) had longer poly(A) tails in livers of *Cnot1-LKO* mice (Fig 3D). Poly(A) tail lengths of *Cxcl10* mRNA in the livers of *Cnot1-LKO* mice were similar to those in controls, although the band intensity increased, suggesting that some mRNAs responsible for "immune system process" increase, regardless of poly(A) elongation (Fig S2B).

### mRNAs preferentially bound by the CCR4–NOT complex are maintained at low levels, at least in part, because of mRNA decay in liver

Total RNA for RNA-seq was prepared from livers of *Cnot1-LKO* mice 14 d after tamoxifen administration, when they were about to die (Fig 1C). Although GO analysis clearly explained the severely damaged livers in *Cnot1-LKO* mice, the data do not necessarily imply a direct CNOT1 effect in liver. To examine whether TF-, cell cycle-, and DNA damage-mRNAs, which increased in livers from *Cnot1-LKO* mice, are controlled by the CCR4–NOT complex in the liver, we conducted RNA immunoprecipitation (RIP) followed by RNA-seq (RIP-seq). When we performed immunoprecipitation using anti-CNOT3 antibody, other subunits of the CCR4–NOT complex were efficiently co-purified (Fig S3A). Gel filtration chromatography showed that CNOT3, as well as the other complex subunits, were present in the same molecular weight fractions, suggesting that almost all CNOT3 existed as a component of CCR4–NOT complex in the liver (Fig S3B). We, thus, reasoned that RIP using anti-CNOT3 antibody (CNOT3-IP) represented levels of CCR4–NOT–RIP. To examine the relationship between the expression level and binding of the CCR4–NOT complex to each mRNA, we normalized *gene* FPKMs

**Table 2.  Serum profiles of control and *Cnot1-LKO* mice.**

| | Cnot1 (fl/fl) (n = 3) | Cnot1-LKO (n = 3) | P-value |
|---|---|---|---|
| AST (IU/L) | 3.17E+02 ± 44.6 | 5.26E+03 ± 1.5E+03 | 9.36E−03 |
| ALT (IU/L) | 3.60E+01 ± 1.15 | 9.48E+03 ± 2.64E+03 | 2.33E−02 |
| ALP (IU/L) | 1.16E+03 ± 88.8 | 1.06E+04 ± 1.04E+03 | 8.16E−04 |
| LDH (IU/L) | 1.01E+03 ± 73.6 | 6.94E+03 ± 8.32E+02 | 2.08E−03 |

Values represent means ± SEM.
Statistical significance was determined by *t* test.

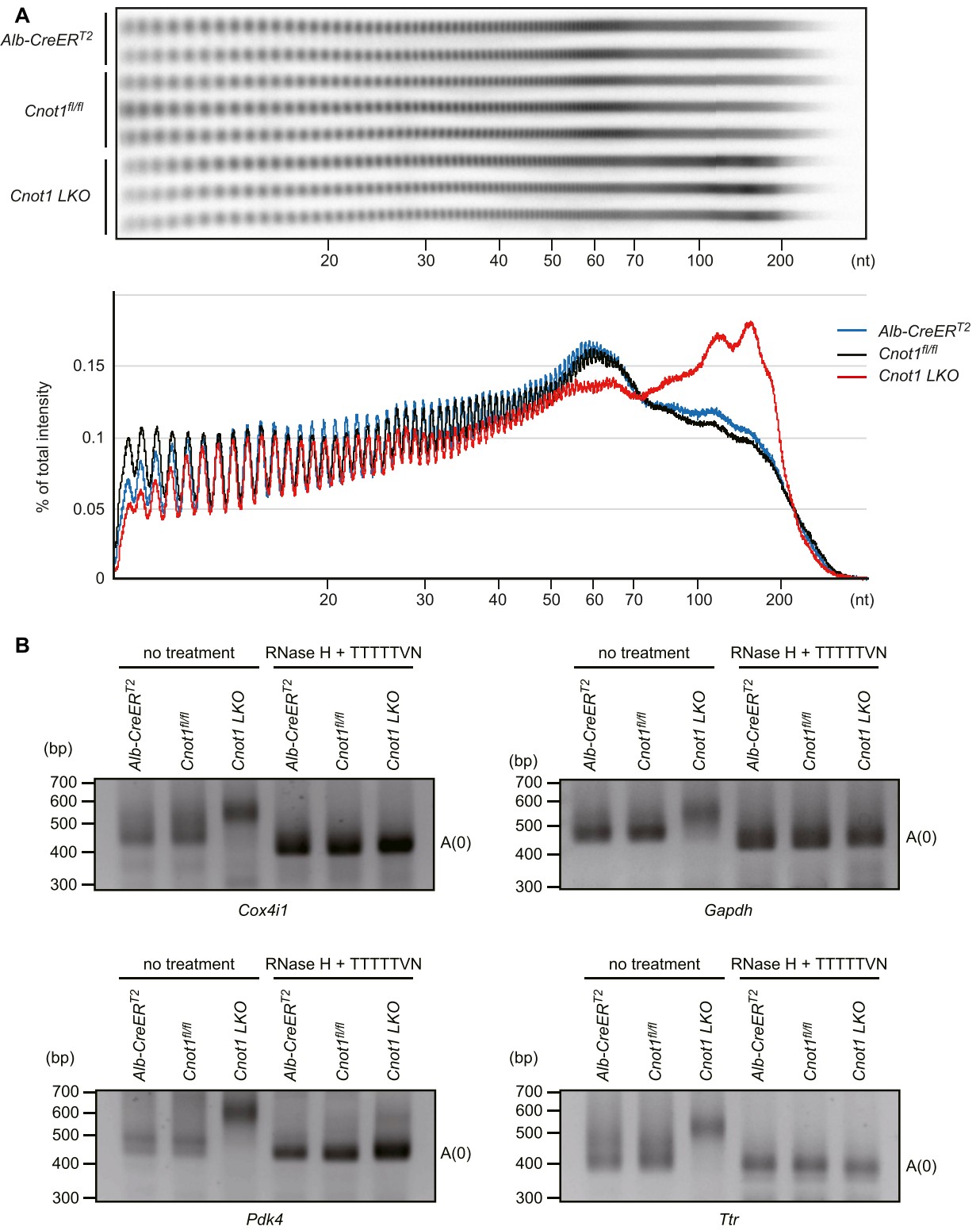

**Figure 2. Elongated poly(A) tails of RNAs in livers from *Cnot1-LKO* mice.**
**(A)** Poly(A) tail lengths of bulk RNA in livers from *Alb-CreER*[T2], *Cnot1*[fl/fl], and *Cnot1-LKO* mice. The lower graph shows a densitogram of poly(A) tail lengths in each genotype. Signal intensity was normalized to total intensity (%). Values represent the mean of independent experiments (*Alb-CreER*[T2]; n = 2, *Cnot1*[fl/fl] and *Cnot1-LKO* mice; n = 3). **(B)** Poly(A) tail lengths of the indicated mRNAs in livers from *Alb-CreER*[T2], *Cnot1*[fl/fl], and *Cnot1-LKO* mice. PCR products of RNAs treated with RNase H in the presence of oligo (dT) primer, which indicates that fragments without poly(A) tails were also loaded.

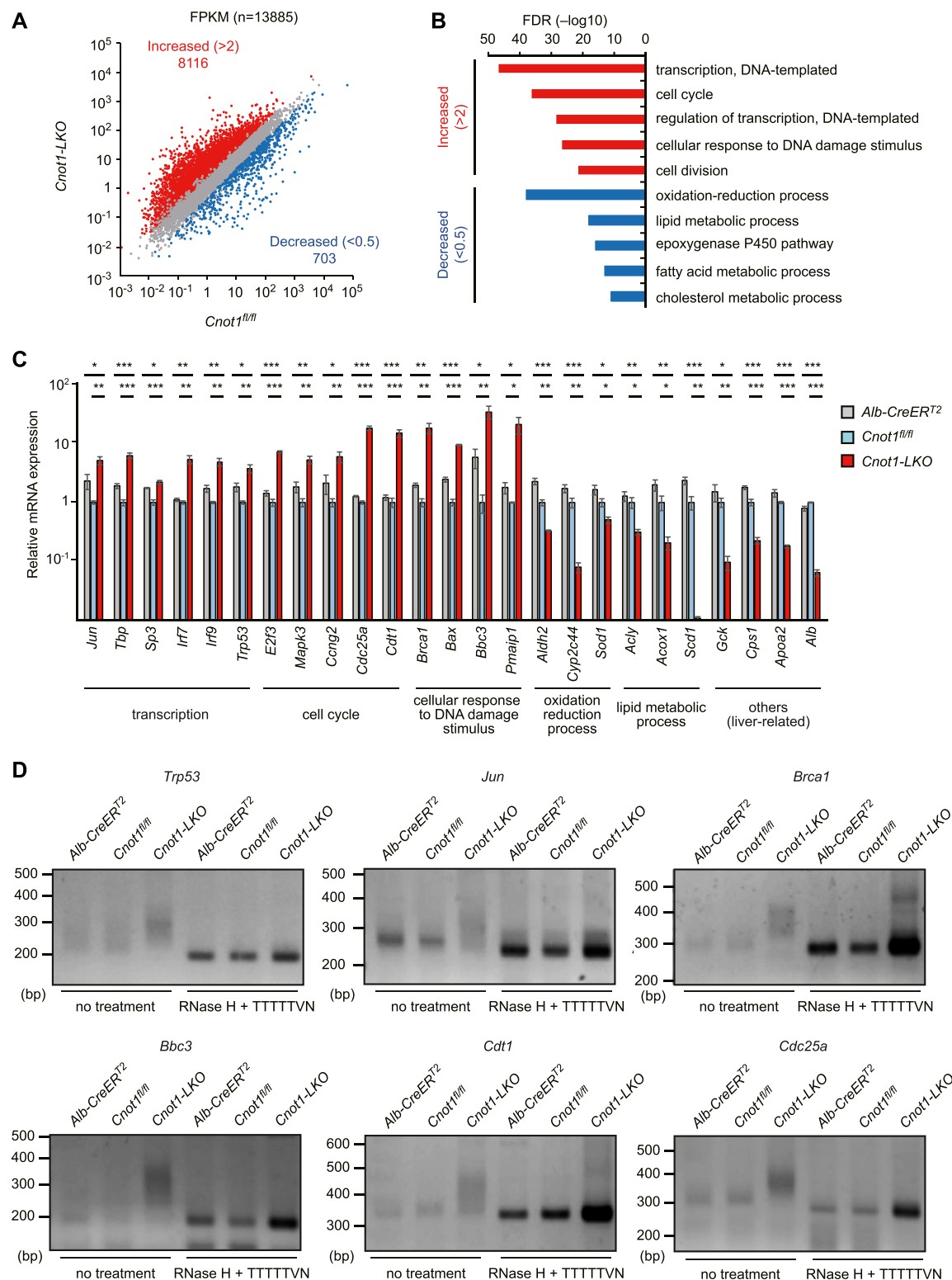

**Figure 3. Increase of TF-, cell cycle-, and DNA damage-mRNAs and decrease in liver function–related mRNAs in livers from *Cnot1-LKO* mice.**
**(A)** Scatterplot of mRNA FPKMs in livers from *Cnot1*<sup>fl/fl</sup> and *Cnot1-LKO* mice. Each dot represents the mean of four independent samples. mRNAs that increased or decreased >2-fold in livers from *Cnot1-LKO* mice compared with those from *Cnot1*<sup>fl/fl</sup> mice are shown in red or blue, respectively (Mann–Whitney U-test, false discovery rate < 0.05). **(B)** Enriched GO categories in which mRNAs increased or decreased >2-fold in livers from *Cnot1-LKO* mice compared with those from *Cnot1*<sup>fl/fl</sup> mice. **(C)** qRT-PCR analysis of the indicated mRNAs in livers from *Alb-CreER*<sup>T2</sup>, *Cnot1*<sup>fl/fl</sup>, and *Cnot1-LKO* mice (n = 4). The *Rplp0* mRNA level was used for normalization. Values in graphs represent means ± SEM. Unpaired *t* test, *$P$ < 0.05, **$P$ < 0.01, ***$P$ < 0.001. **(D)** Poly(A) tail lengths of the indicated mRNAs in livers from *Alb-CreER*<sup>T2</sup>, *Cnot1*<sup>fl/fl</sup>, and *Cnot1-LKO* mice. PCR products of RNAs treated with RNase H in the presence of oligo (dT) primer, which indicates that fragments without poly(A) tails were also loaded.

in RIP using *gene* FPKMs in input total RNA (defined as the CCR4–NOT–RIP enrichment value). TF-, cell cycle-, and DNA damage-mRNAs showed higher CCR4–NOT–RIP enrichment values than liver function–related mRNAs (Fig 4A and B and Table S3). Binding of the CCR4–NOT complex to those mRNA species was confirmed with qRT-PCR of CCR4–NOT–RIP samples from the whole liver (Fig S3C). We obtained similar qRT-PCR results with TF-, cell cycle-, and DNA damage-mRNAs after CCR4–NOT–RIP using isolated hepatocytes (Fig S3D), indicating that the CCR4–NOT complex controls the levels of these mRNAs in hepatocytes. On the other hand, binding of the CCR4–NOT complex to *Cxcl10* and *Tlr3* mRNAs was not significant in isolated hepatocytes (Fig S3D). Therefore, it is possible that some mRNAs detected in CCR4–NOT–RIP using whole liver lysates, in particular "immune system process"–related mRNAs, are from cells other than hepatocytes. When we ranked mRNA species according to their FPKMs in liver, TF-, cell cycle-, and DNA damage-mRNAs generally belonged to the lower expression group compared with liver function–related mRNAs (Fig 4C and D and Table S4). Furthermore, mRNAs expressed at lower levels in the liver had higher CCR4–NOT–RIP enrichment values (Fig 4E). These findings suggest that the CCR4–NOT complex binds to more mRNA species that are expressed at low levels in the liver, such as TF-, cell cycle-, and DNA damage-mRNAs, than to mRNA species that are expressed at high levels in the same tissue, such as liver function–related mRNAs. Similar results were obtained when we used the medians of both RIP enrichment values and FPKMs, instead of the means (Fig S4).

We next examined whether binding of the CCR4–NOT complex to mRNAs influences their decay rates in the liver. We performed chase experiments by injecting the transcription inhibitor, actinomycin D (Act. D), into mice. Total RNAs were prepared from Act. D–injected mouse livers and were subjected to RNA-seq. We calculated mRNA half-lives using the RNA-seq results (see the Materials and Methods section). Because levels of many mRNA species decrease as a result of mRNA decay in Act. D–treated samples, normalization with a level of stable mRNA will be more effective for calculation of mRNA half-lives than genome-wide normalization methods. We normalized *gene* FPKMs with the FPKM of *60S acidic ribosomal protein P0* (*Rplp0*) mRNA, which was stable during 8 h Act. D treatment (Fig S5). We compared the calculated mRNA half-lives in this study with previously reported data from chase experiments (Friedel et al, 2009; Sharova et al, 2009; Schwanhausser et al, 2011). Although the data were from NIH3T3 and mouse embryonic stem cells, there was weak, but significant correlation between our data and the others, suggesting that Act. D chase experiments using mice worked properly (Fig S6A). We found that TF-, cell cycle-, and DNA damage-mRNAs have relatively shorter mRNA half-lives than liver function–related mRNAs (Fig 4F and G and Table S5). Overall, mRNAs that have shorter half-lives showed higher CCR4–NOT–RIP enrichment values and lower expression (Fig 4H and I). These data suggest that TF-, cell cycle-, and DNA damage-mRNAs are targeted for degradation and are restricted to relatively low expression levels in the liver in a CCR4–NOT complex–dependent manner. In contrast, mRNA species that are expressed at high levels in the liver seem to escape binding of the CCR4–NOT complex and subsequent decay.

## Elongation of mRNA half-lives in livers from *Cnot1-LKO* mice

By comparing mRNA half-lives in the livers of control and *Cnot1-LKO* mice, we found that 6,718 mRNAs, had longer half-lives (*Cnot1-LKO*/control >2.0) and 187 mRNAs had shorter half-lives (*Cnot1-LKO*/control <0.5) in the livers from *Cnot1-LKO* mice (Fig 5A and Table S6). mRNAs in the GO categories "transcription," "cell cycle," and "cellular response to DNA damage stimulus" had significantly longer half-lives in the livers of *Cnot1-LKO* mice than in those of control mice (Figs 5B and S6B). We found that stabilized mRNAs had higher CCR4–NOT–RIP enrichment values than mRNAs that were destabilized or unchanged (half-lives, 0.5< *Cnot1-LKO*/control <2.0) in the livers from *Cnot1-LKO* mice compared with control mice (Fig 5C). Furthermore, CCR4–NOT–RIP enrichment values correlated significantly with the increase in gene FPKMs and mRNA half-lives in livers (*Cnot1-LKO*/control) (Fig 5D and E). Therefore, CCR4–NOT–RIP enrichment values basically represent the dependency of mRNAs on CCR4–NOT complex–mediated decay. It should be noted that mRNAs involved in "oxidation–reduction process" and "lipid metabolic process" had longer half-lives in the livers from *Cnot1-LKO* mice (Fig 5B). These data suggest that mRNAs displaying low CCR4–NOT–RIP enrichment values in the liver are also controlled by the CCR4–NOT complex, although loss of control did not necessarily lead to an increase in their expression levels.

## The CCR4–NOT complex destabilizes mRNAs mainly through BRF1 and AGO2 in the liver

We next sought to determine how those mRNAs are targeted by the CCR4–NOT complex. As BRF1 or AGO2 interacts with the CCR4–NOT complex (Fabian et al, 2011; Adachi et al, 2014), we conducted RIP-seq using antibodies against BRF1 and AGO2 (Tables S7 and S8). Immunoprecipitated BRF1 and AGO2 were verified by immunoblot analysis (Fig 6A). We analyzed RNA-seq data of BRF1–RIP and found that 25 of the top 30 enriched mRNAs in BRF1–RIP had consensus AU-rich motifs. Those are not enriched in the control IgG-IP (Table S7). Furthermore, in BRF1–RIP, only 33 mRNAs were included among the top 1,500 enriched mRNAs in control IgG-IP. *miR-122* is an abundant liver miRNA that accounts for 70% of liver total miRNAs. We found that many *miR-122* targets, such as *Aldoa*, *Map3k1*, *Ndrg3*, and *Bcl9* mRNAs, were enriched in AGO2–RIP (Tsai et al, 2012; Luna et al, 2017 and Table S8). Again, they were not enriched in control IgG-IP. These data suggest that both BRF1–RIP and AGO2–RIP worked properly. Scatterplots showed that CCR4–NOT–RIP enrichment values were significantly correlated with BRF1–RIP or AGO2–RIP enrichment values (Fig 6B and C). Consistent with this, mRNAs that showed high BRF1–RIP or AGO2–RIP enrichment values (>1.5) were included in mRNA groups showing relatively high CCR4–NOT–RIP enrichment values (Fig 6D). Furthermore, many mRNAs that were stabilized (half-lives, *Cnot1-LKO*/control >2.0) in the livers from *Cnot1-LKO* mice were included in mRNA groups showing high BRF1–RIP or AGO2–RIP enrichment values (Fig 6E and F). In total, 3,589 species of mRNA were stabilized in the livers from *Cnot1-LKO* mice (half-lives, *Cnot1-LKO*/control >2.0) and largely enriched in CCR4–NOT–RIP (enrichment >1.5). Around half of 3,589 mRNAs were common in those enriched in both BRF1–RIP and AGO2–RIP (Fig 6G). These results suggest that the CCR4–NOT complex promotes

Figure 4. **The CCR4–NOT complex preferentially binds to TF-, cell cycle-, and DNA damage-mRNAs and maintains their expression at low levels.**
**(A)** mRNAs were ordered according to their CCR4–NOT-RIP enrichment values. The x-axis represents ranking in ascending order. The CCR4–NOT-RIP enrichment value represents gene FPKMs in RNA included in the anti-CNOT3 immunoprecipitates normalized against gene FPKMs in the liver total RNA (Input). Means of the values in three independent experiments were used. Representative mRNAs possessing specific functions are shown in red (cell cycle), black (transcription), green (DNA damage response), and blue (metabolism). **(A, B)** Violin plot of CCR4–NOT-RIP enrichment values calculated in (A) for all mRNAs, and those categorized in the indicated GO terms. **(C)** mRNAs were ordered according to their FPKMs in livers from control mice (means of the values in four mice). **(A)** The x-axis represents ranking in ascending

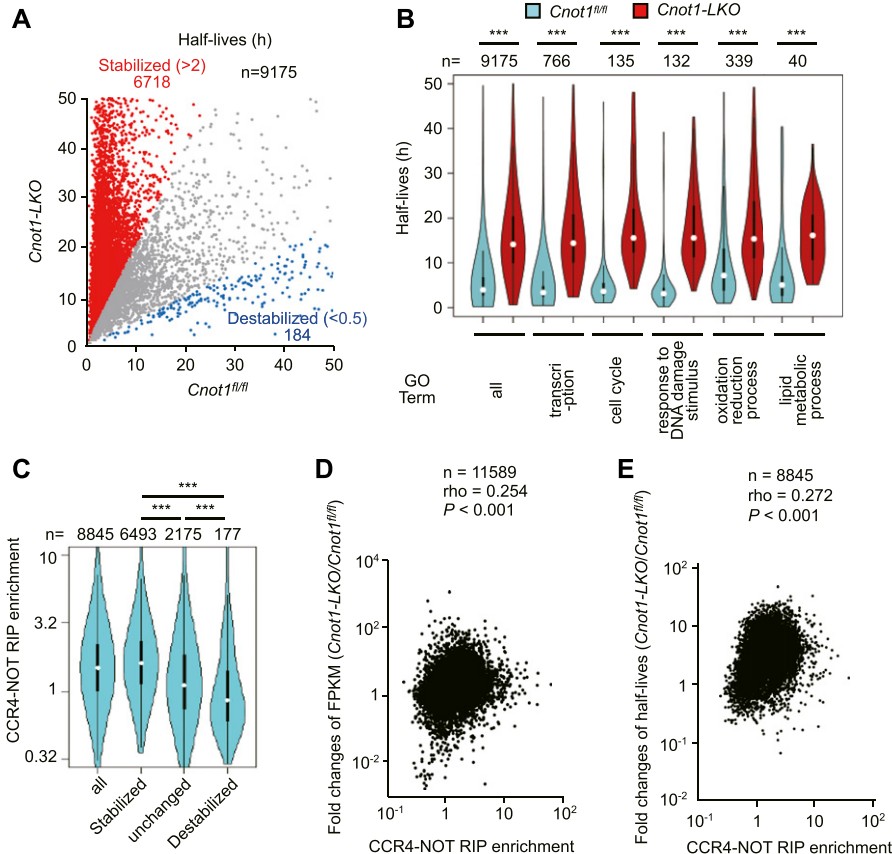

**Figure 5. Elongated half-lives of mRNAs in livers from *Cnot1-LKO* mice, and correlation between mRNA stabilization and RIP enrichment.**

**(A)** Scatterplot of mRNA half-lives in livers from control (*Cnot1^{fl/fl}*) and *Cnot1-LKO* mice. Calculation of mRNA half-lives in livers from *Cnot1-LKO* mice was performed as in Fig 4F (see the Materials and Methods section). mRNAs with half-lives elongated or shortened in livers from *Cnot1-LKO* mice by more than twofold compared with those from *Cnot1^{fl/fl}* mice, are shown in red or blue, respectively. **(B)** Violin plot of half-lives for all mRNAs and those categorized in the indicated GO terms in the livers from control (*Cnot1^{fl/fl}*) and *Cnot1-LKO* mice. **(C)** Violin plot of CCR4–NOT–RIP enrichment values in livers from control (*Cnot1^{fl/fl}*) mice (means of three independent experiments) for all mRNAs, stabilized, destabilized, and unchanged mRNAs (changes in half-lives: *Cnot1-LKO*/control >2.0, <0.5, and the others, respectively). **(D)** Scatterplot of CCR4–NOT–RIP enrichment values and changes in FPKMs in livers (*Cnot1-LKO*/control). **(E)** Scatterplot of CCR4–NOT–RIP enrichment values and changes of mRNA half-lives in livers (*Cnot1-LKO*/control). **(D, E)** Spearman's rank correlation coefficient (rho) and *P*-value were calculated (D, E). **(B, C)** Wilcoxon signed-rank test (B) and Wilcoxon rank sum test (C) were used. *$P < 0.05$, **$P < 0.01$, ***$P < 0.001$.

degradation of a variety of mRNAs, mainly through BRF1- or AGO2-mediated target recognition in the liver. Finally, we found that sequences enriched with U or A; "TTTTGT T/G T" and "TTTTTAAA" were frequently observed in the 3′-UTRs of stabilized, but not destabilized mRNAs in the livers from *Cnot1-LKO* mice (Fig S7).

### Differential expression of pre-mRNAs in the livers between control and *Cnot1-LKO* mice

In the livers from *Cnot1-LKO* mice, more than 80% of the mRNAs that we analyzed showed both elongated half-lives and increased expression compared with control livers (Fig 7A). On the other hand, around 10% of the mRNAs expressed at lower levels in the livers from *Cnot1-LKO* mice than in controls had elongated half-lives unexpectedly (Fig 7A). As steady-state mRNA levels are determined by both mRNA synthesis and degradation, we examined whether the transcription state was altered in the livers from *Cnot1-LKO* mice. Previous reports showed that changes in pre-mRNA levels are

significantly correlated with transcription rates (Gaidatzis et al, 2015; Wang et al, 2018). We performed comprehensive profiling of pre-mRNAs by counting the number of intron sequence reads (intron reads) using RNA-seq data in the livers from control and *Cnot1-LKO* mice (Table S9). The mRNA synthesis rate divided by the mRNA decay rate is defined as the steady-state level of mRNA (Palumbo et al, 2015), and the mRNA decay rate is related to the reciprocal of the half-life (Chen et al, 2008). By considering pre-mRNA levels as the mRNA synthesis rate, mRNA expression level could be estimated by multiplying mRNA half-lives and pre-mRNA levels. Calculated values corresponded well to mRNA expression values obtained from RNA-seq, further suggesting that pre-mRNA levels are good proxies for transcription rates (Fig S8A).

The landscape of intron reads was similar to that of exon counts (compare Fig S8B and C with Figs 3A and S2A), although there was no statistical significance in differences of pre-mRNA expression levels between control and *Cnot1-LKO* mice. To determine whether specific pre-mRNAs were differentially expressed in the livers of control and

order. Representative mRNAs are shown as in (A). **(B, D)** Violin plot of FPKMs for all mRNAs and grouped mRNAs as in (B). **(E)** Scatterplot of CCR4–NOT–RIP enrichment values and mRNA FPKMs in livers from control mice. **(F)** Calculation of mRNA half-lives using RNA-seq results of liver total RNAs, which were prepared from Act. D–injected control (*Cnot1^{fl/fl}*) mice, are described in the Materials and Methods section. mRNAs were ordered according to lengths of their half-lives. The x-axis represents ranking in ascending order. **(A)** Representative mRNAs are shown as in (A). **(B, F, G)** Violin plot of mRNA half-lives calculated in (F) for all mRNAs and grouped mRNAs as in (B). **(H)** Scatterplot of CCR4–NOT–RIP enrichment values and mRNA half-lives in livers from control mice. **(I)** Scatterplot of mRNA half-lives and FPKMs in livers from control mice. **(E, H, I)** Spearman's rank correlation coefficients (rho) and *P*-values were calculated (E, H, I). **(A, F)** Note that representative mRNAs missing in (A) and (F) did not satisfy criteria for the analyses (see the Materials and Methods section).

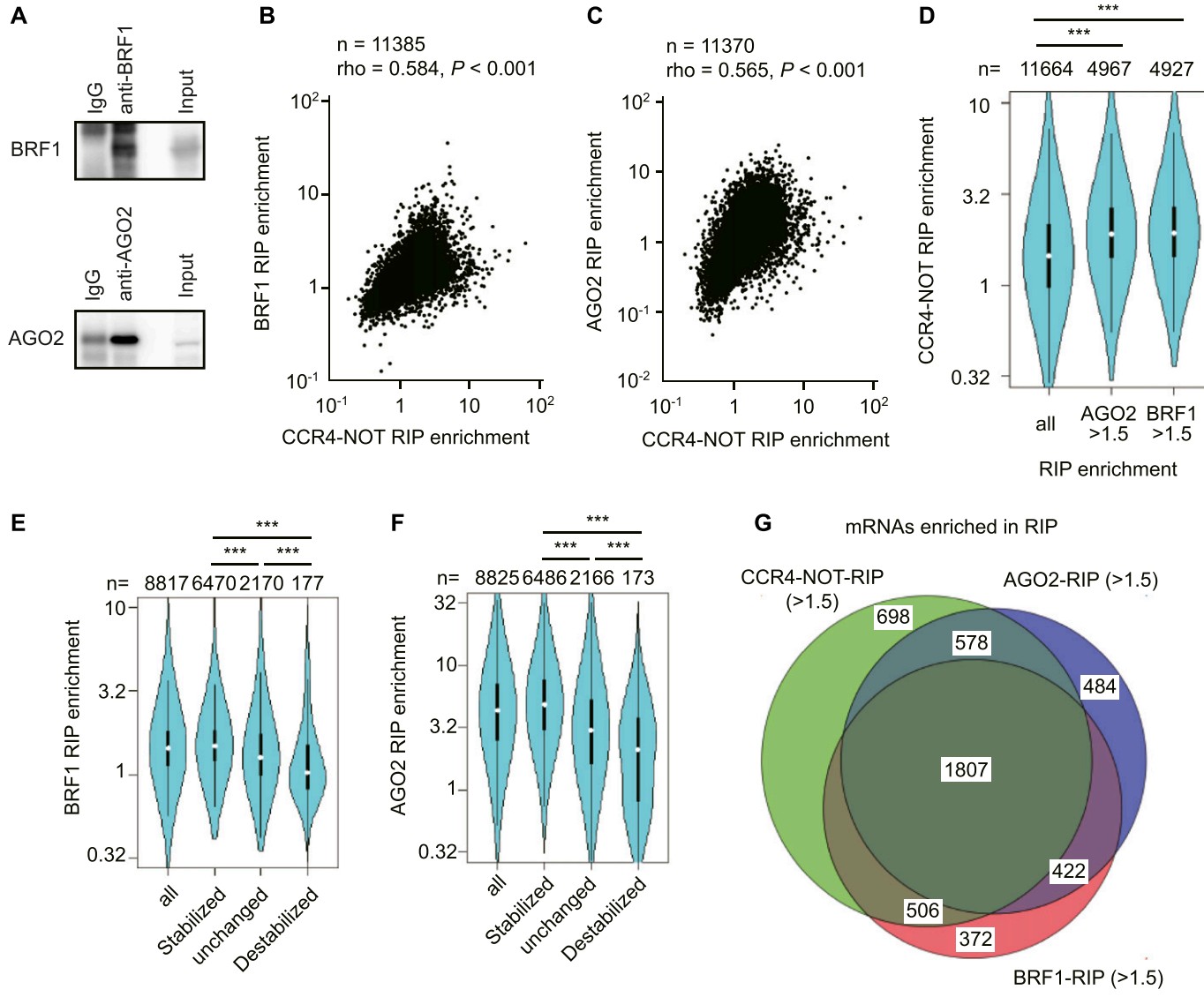

**Figure 6. The CCR4–NOT complex destabilizes mRNAs mainly through BRF1 and AGO2 in liver.**
**(A)** Immunoblot analysis of anti-BRF1 (upper) or anti-AGO2 (lower) immunoprecipitates that were used for RIP-seq. **(B, C)** Scatterplots of CCR4–NOT–RIP enrichment values versus BRF1–RIP enrichment values (BRF1–RIP FPKM/Input FPKM) (B), or versus AGO2–RIP enrichment values (AGO2–RIP FPKM/Input FPKM) (C) in livers from control mice. **(D)** Violin plot of CCR4–NOT–RIP enrichment values in all mRNAs, BRF1-bound mRNAs (BRF1–RIP enrichment values > 1.5), and AGO2-bound mRNAs (AGO2–RIP enrichment values > 1.5) in livers from control mice. **(E, F)** Violin plots of BRF1–RIP (E) or AGO2–RIP (F) enrichment values for all mRNAs and grouped mRNAs, as in Fig 5C. **(G)** Venn diagram of mRNAs showing elongated half-lives in livers from *Cnot1-LKO* mice (changes in half-lives: *Cnot1-LKO*/control >2.0). mRNAs with RIP enrichment values more than 1.5 in CCR4–NOT–RIP (green), BRF1–RIP (red), and AGO2–RIP (blue) were compared. **(B, C, D, E, F)** Means of values in three independent experiments were used (B, C, D, E, F). **(B, C)** Spearman's rank correlation coefficient (rho) and the *P*-value were calculated (B, C). **(D, E, F)** Wilcoxon rank sum test (D, E, F), *P < 0.05, **P < 0.01, ***P < 0.001.

*Cnot1-LKO* mice, we performed qRT-PCR analysis using intron region–specific primers. The results showed that pre-mRNA levels of "immune system process"–related genes (*Mx1* and *Cxcl10*) and "apoptosis"-related genes (*Ripk3*, *Pmaip1*, and *Bax*) increased in the livers of *Cnot1-LKO* mice (Fig 7B). Pre-mRNA levels of genes encoding TFs (*Trp53*, *Sp3*, *Tbp*, and *Irf9*) and cell cycle-related genes (*Cdt1* and *Cdc25a*) were comparable in the livers of control and *Cnot1-LKO* mice (Fig 7C). We found that binding motifs for interferon regulatory factor (IRF) proteins or Trp53 were enriched in the promoter regions of "immune system process"–related genes or

"apoptosis"-related genes, which increased in the livers from *Cnot1-LKO* mice at pre-mRNA levels, respectively (Fig S9). To examine whether the increase in pre-mRNA levels was relevant to transcriptional activation, we conducted chromatin immunoprecipitation assay (ChIP) using an antibody against RNA polymerase II. The results showed that RNA polymerase II occupancy on genomic regions of "immune system process"–related genes (*Ccl2* and *Cxcl10*) and "apoptosis"-related genes (*Ripk3*, *Pmaip1*, and *Bax*) increased in the livers from *Cnot1-LKO* mice (Fig 7D). Therefore, it is possible that Trp53 and IRF9 proteins increased

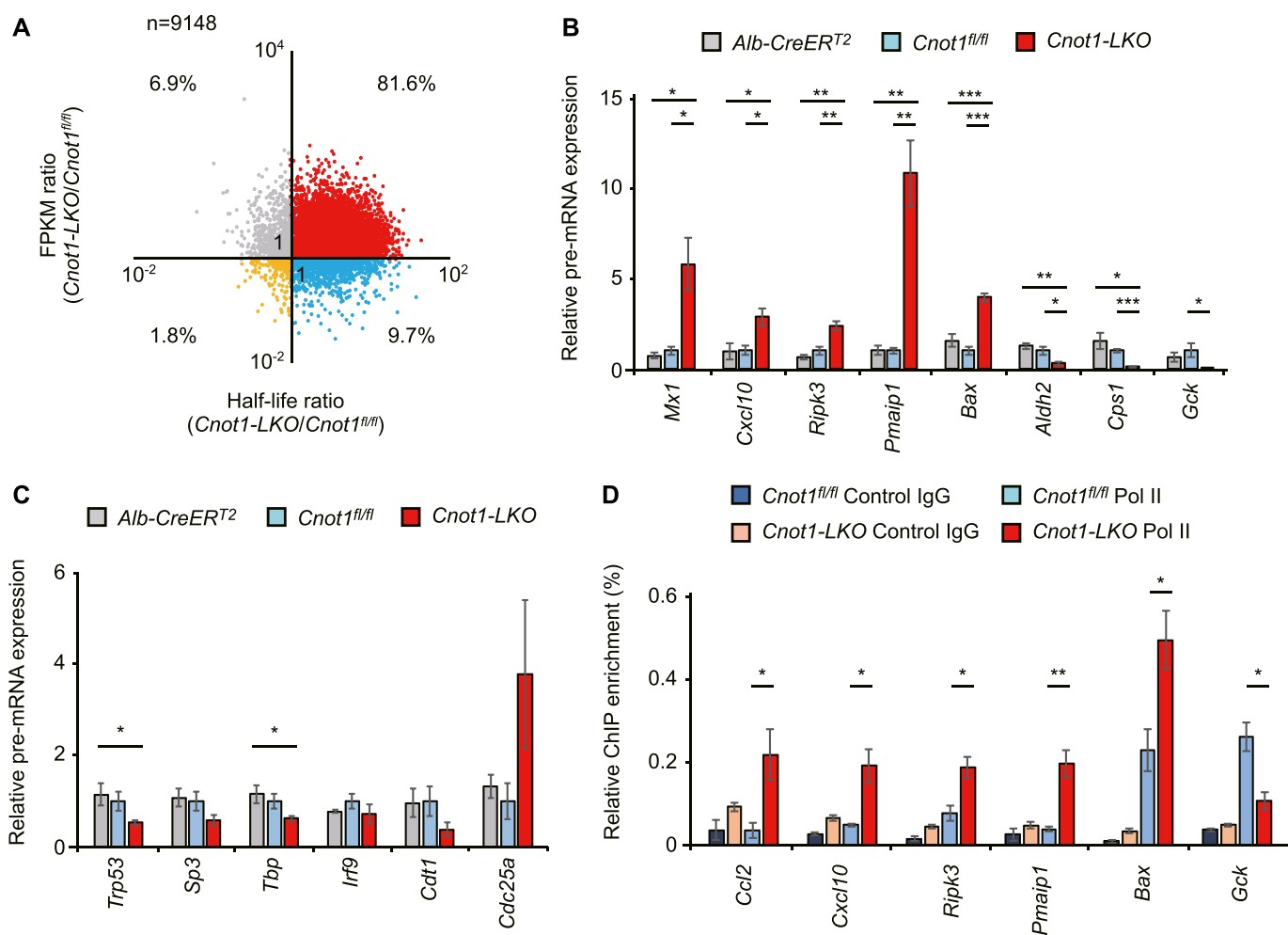

**Figure 7.  Increase in immune system process- and apoptosis-genes and decrease in liver-related genes at pre-RNA levels in livers from *Cnot1-LKO* mice partly reflect changes in transcription.**
**(A)** Scatterplot of changes in mRNA half-lives and FPKMs (*Cnot1-LKO*/control). **(B, C)** qRT-PCR analysis of the indicated pre-mRNAs in livers from control (*Alb-CreER^{T2}* and *Cnot1^{fl/fl}*) and *Cnot1-LKO* mice (n = 4). Pre-mRNA levels were normalized with the *Rplp0* pre-mRNA level. **(D)** ChIP-assay using the antibody against RNA polymerase II (Pol II) or control IgG. qRT-PCR analysis of co-immunoprecipitated genome DNA fragments in livers from control (*Cnot1^{fl/fl}*) and *Cnot1-LKO* mice was performed using primers in genomic regions of the indicated genes. Percentages against input genome DNA were calculated (n = 3). Values in graphs represent means ± SEM. Unpaired *t* test, *$P < 0.05$, **$P < 0.01$, ***$P < 0.001$.

because of mRNA stabilization and subsequently induced the transcription of "immune system process"–related and "apoptosis"-related genes in the livers from *Cnot1-LKO* mice. This could partly explain why *Cxcl10* mRNA increased without poly(A) elongation in *Cnot1-LKO* mice (Fig S2B). It is also possible that infiltration of immune cells into the livers of *Cnot1-LKO* mice contributes to the increase (Fig 1). On the other hand, pre-mRNA levels of liver function–related genes (*Aldh2*, *Cps1*, and *Gck*) significantly decreased in the livers from *Cnot1-LKO* mice (Fig 7B). Therefore, expression of some genes involved in "oxidation–reduction process" and "lipid metabolic process" decreased at pre-mRNA levels in the livers from *Cnot1-LKO* mice. RNA polymerase II occupancy on the genomic region of the *Gck* gene decreased significantly in the livers from *Cnot1-LKO* mice compared with control mice (Fig 7D). These results suggest that decreased transcription of some liver function–related genes overwhelmed mRNA stabilization, resulting in decreases of their mRNA levels.

## Indirect effects of *Cnot1* deficiency on the liver transcriptional program

Previous studies have shown that the CCR4–NOT complex exists in the nucleus and directly facilitates transcription in yeast and certain types of mammalian cells (Collart & Struhl, 1994; Badarinarayana et al, 2000; Hu et al, 2009; Kruk et al, 2011; Miller & Reese, 2012; Cejas et al, 2017). We examined subcellular localization of subunits of the CCR4–NOT complex as well as relevant transcription regulators in the mouse liver. CNOT1 and CNOT2 were localized in the cytoplasm but were barely detectable in the nucleus (Fig 8A and B). Although Trp53 and TBP were expressed at very low levels in livers from control mice, they were significantly elevated in both the nuclei and cytoplasm in the livers from *Cnot1-LKO* mice (Fig 8A and B). RPB1, a component of RNA polymerase II, and IRF9 increased significantly in the livers from *Cnot1-LKO*

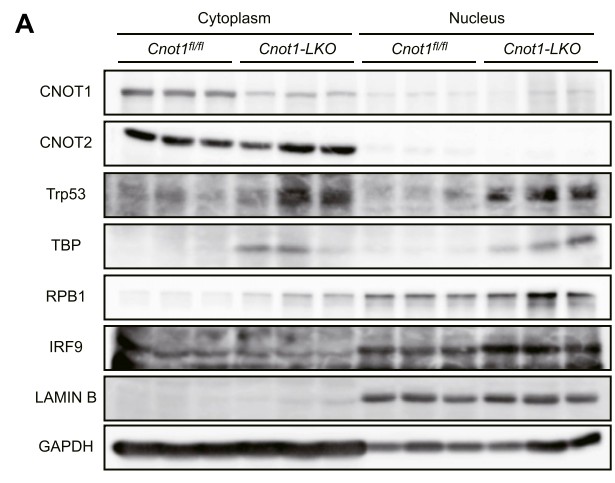

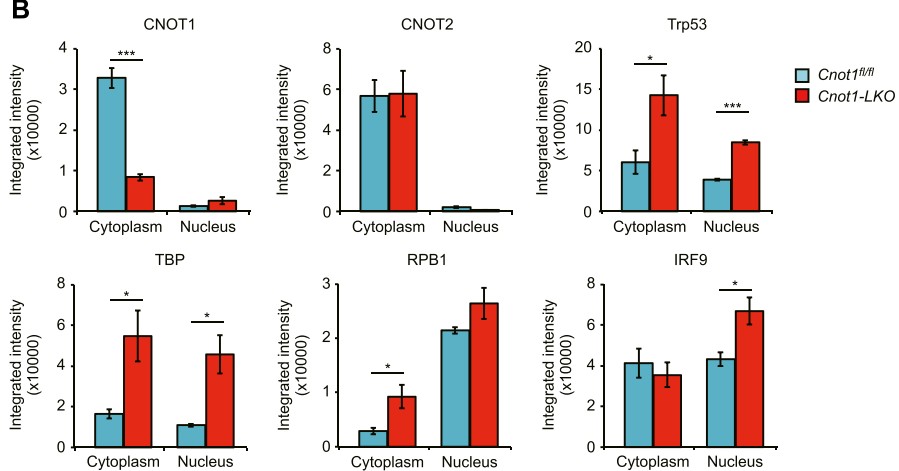

**Figure 8. Cytoplasmic localization of CCR4–NOT complex subunits.**
**(A)** Immunoblot analysis of cytoplasmic and nuclear fractions isolated from livers of control (*Cnot1fl/fl*) and *Cnot1-LKO* mice using the indicated antibodies (n = 3). **(A, B)** Quantification of the immunoblot results in (A). Proteins in the cytoplasmic and nuclear fractions were normalized with GAPDH and LAMIN B, respectively. Values in graphs represent means ± SEM. Unpaired *t* test, *P < 0.05, ***P < 0.001.

mice, in the nuclear or cytoplasmic fraction, respectively (Fig 8A and B). Consistent with the cytoplasmic localization of CNOT1 and CNOT2, we did not detect interaction of the CCR4–NOT complex with RPB1 and TBP (Fig S3A). Taken together, these data suggest that the CCR4–NOT complex does not directly influence transcription in the liver and that impaired mRNA degradation in the absence of CNOT1 secondarily influences transcription or other mRNA regulatory processes.

## Discussion

In this study, we provide evidence that the CCR4–NOT deadenylase complex plays critical roles in liver homeostasis. Liver-specific disruption of *Cnot1*, which encodes a scaffold subunit of the complex, resulted in substantial elongation of bulk RNA poly(A) tails, as in the case of *Drosophila* S2 cells, *Caenorhabditis elegans*, and MEFs (Temme et al, 2010; Nousch et al, 2013; Mostafa et al, 2020), further indicating an essential role of CNOT1 in CCR4–NOT complex–mediated mRNA deadenylation in vivo. Consistent with previous reports that poly(A) tails stabilize mRNAs in eukaryotes (Dreyfus & Regnier, 2002; Weill et al, 2012), more than 80% of the

mRNAs that we analyzed showed elongated half-lives in livers from *Cnot1-LKO* mice (Fig 7A). Therefore, the CCR4–NOT complex is the major mRNA decay mechanism promoting mRNA turnover in the liver.

Short mRNA half-lives correlated with reduced mRNA levels in HeLa cells and mouse liver (Maekawa et al, 2015, and this study). This suggests that mRNA decay mechanisms help maintain low expression of some mRNAs. This study showed that TF-, cell cycle- and DNA damage-mRNAs were mainly restricted to low expression by the CCR4–NOT complex in the liver (Figs 3 and 4). This role of the CCR4–NOT complex is important in liver homeostasis because excess cell cycle- and DNA damage-mRNAs are unfavorable for cells and mature tissues. Indeed, suppression of the CCR4–NOT complex altered cellular homeostasis, tissue development, and tissue function (Aslam et al, 2009; Mittal et al, 2011; Inoue et al, 2015; Yamaguchi et al, 2018; Suzuki et al, 2019). Importantly, TF-mRNAs have significantly shorter half-lives compared with other mRNA species and undergo rapid decay in several cell lines, including hepatic cells (Yang et al, 2003), suggesting that regulation of TF-mRNAs by the CCR4–NOT complex also contributes to liver homeostasis. On the other hand, liver function–related mRNAs were only moderately controlled by the CCR4–NOT complex. Those mRNAs are generally active in translation to ensure liver function.

We hypothesize that they are protected from mRNA decay because active translation status contributes to mRNA stabilization and suppression of translation repression is coupled with mRNA degradation (Hendrickson et al, 2009; Guo et al, 2010; Edri et al, 2014). Taken together, by facilitating shortening of poly(A) tail lengths of mRNAs, the CCR4–NOT complex maintains liver health.

Several lines of evidence show that the CCR4–NOT complex directly regulates transcription. In *Saccharomyces cerevisiae*, the Ccr4–Not complex interacts with Tbp and Taf proteins to suppress transcription initiation (Collart & Struhl, 1994; Badarinarayana et al, 2000; Miller & Reese, 2012), whereas it stimulates transcription elongation through interaction with RNA polymerase II (Kruk et al, 2011). In mouse embryonic stem cells and progressive cancer cell lines, the CCR4–NOT complex is localized in the nucleus and binds to the promoter region of self-renewal genes (Hu et al, 2009; Cejas et al, 2017). The CCR4–NOT complex interacts with nuclear receptors, such as RXR and ERα, and regulates transcription of their target genes in a ligand-dependent manner (Nakamura et al, 2004; Winkler et al, 2006; Garapaty et al, 2008). Furthermore, the CCR4–NOT complex binds to the TF, EBF1, and modifies B-cell differentiation (Yang et al, 2016). However, we hardly detected the CCR4–NOT complex in liver nuclear fractions, suggesting indirect involvement of the CCR4–NOT complex in transcription in the liver (Fig 8). One possibility is that stabilization of TF-mRNAs and subsequent increases of the proteins influences transcriptional programs. Especially, Trp53- and IRF-induced transcription appears to be involved because many genes that increased in the livers from *Cnot1-LKO* mice at pre-mRNA levels had binding elements for those TFs in their promoter regions (Fig S9). Indeed, Trp53-dependent transcriptional activation has considerable effects on abnormalities observed upon suppression of the CCR4–NOT complex in B-cells and heart (Inoue et al, 2015; Yamaguchi et al, 2018). RNA polymerase II loading on a genomic region of the *Gck* gene decreased in the livers from *Cnot1-LKO* mice (Fig 7D). When transcription of DNA damage response- or immune system process genes is extensively induced, transcription of liver function–related genes may be limited. Decreases in mRNA decay rates in *Cnot1-LKO* mice might be balanced by adjustments in mRNA synthesis rate. It will be interesting to determine whether transcript buffering occurs in the livers of *Cnot1-LKO* mice because this phenomenon is well established in yeast, but not in other species yet (Timmers & Tora, 2018).

Previous reports showed that changes in pre-mRNA levels are significantly correlated with those in transcription rates (Gaidatzis et al, 2015; Wang et al, 2018). Here, we showed that suppression of the CCR4–NOT complex also influenced the abundance of pre-mRNAs. Consequently, whereas TF-, cell cycle-, and DNA damage-mRNAs increased, liver function–related mRNAs decreased. ChIP experiments imply that different expression levels of pre-mRNAs are due at least in part to changes in transcription rate. We analyzed only some of the genes; thus, we cannot exclude the possibility that other mechanisms such as splicing, nuclear RNA decay, or nuclear RNA export might be responsible for changes in pre-mRNA levels. It is also possible that decreased levels of mRNAs encoding liver function–related molecules are only relative to other mRNA species that increased significantly in livers of *Cnot1-LKO* mice. Further analyses are necessary to clarify the molecular mechanism by which CNOT1 suppression leads to global changes in mRNA expression in the liver.

RIP experiments showed that mRNAs preferentially bound by the CCR4–NOT complex overlap with those targeted by BRF1 and AGO2, suggesting that BRF1 and AGO2 contribute to CCR4–NOT complex–dependent mRNA decay in the liver. Liver-specific suppression of *Ago2* did not induce obvious abnormalities during development or in adulthood (Zhang et al, 2018). Liver-specific *Brf1*-deficient mice displayed abnormal bile acid and lipid metabolism, although an appearance of inflammatory phenotypes was not described (Tarling et al, 2017). Although AGO2 could be compensated because of potentially overlapping roles of other family proteins such as AGO1 (Dueck et al, 2012), our results suggest that a set of mRNAs is redundantly controlled by BRF1 or AGO2. It should be noted that not all mRNAs bound by the CCR4–NOT complex are explained by BRF1- or AGO2-mediated mechanisms (Fig 6G). Other RNA-binding proteins (RBPs) are involved in the regulation of those mRNAs. Indeed, the CCR4–NOT complex uses various RBPs to recognize target mRNAs (Chang et al, 2004; Chicoine et al, 2007; Hosoda et al, 2011; Leppek et al, 2013; Bhandari et al, 2014; Ogami et al, 2014; Sgromo et al, 2017; Yamaji et al, 2017). Our results suggest that RBPs that recognize sequences enriched with U or A may be good candidates (Fig S7).

Fulminant hepatitis causes acute, severe liver injury as a result of massive hepatocyte apoptosis and necrosis induced by death receptor signaling, involving FAS, TNF receptor, and TNFSNF10b, leading to lethality (Malhi et al, 2010). Our liver-specific *Cnot1* disruption model minimized extrahepatic causes, and the phenotype was reproducible and irreversible under our experimental conditions. Thus, *Cnot1-LKO* mice could be added to hepatitis models to help develop therapeutics for fulminant hepatitis. Because the *Cnot1* gene has single nucleotide polymorphisms that are associated with hepatitis C virus infection and hepatic toxicity (Li et al, 2009; Dzikiewicz-Krawczyk, 2015), studies using clinical samples may reveal the relationship between dysfunction of the CCR4–NOT complex and the onset of hepatitis.

# Materials and Methods

### Mice

*Cnot1* conditional KO (*Cnot1^fl/fl*) mice (Accession No CDB0916K: http://www2.clst.riken.jp/arg/mutant%20mice%20list.html) were generated with TT2 ES cell lines (Yagi et al, 1993) as described previously (http://www2.clst.riken.jp/arg/methods.html). To generate conditional alleles (floxed alleles) from targeted alleles, mice with targeted alleles were crossed with mice expressing FLP (#009086; Jackson Laboratory). Wild-type, floxed, and KO alleles were detected with PCR primers: 5′-CCACTGACTTGACACTATTAGTGTGAAAGG-3′ for the forward primer of wild-type, floxed, and KO alleles, 5′-CCAGAGCTGTCTAGGCAGACAAGG-3′ for the reverse primer of wild-type and floxed alleles, and 5′-CCAGGTGCTGACAATACTGAGGATAGTCC-3′ for the reverse primer of a KO allele. PCR product sizes for wild-type, floxed, and KO alleles were 279, 492, and 732 bp, respectively. The absence of FLP knock-in alleles in mice with floxed alleles was also confirmed by PCR. We backcrossed *Cnot1^fl/fl* mice with C57BL/6J mice for at least eight generations. Mice were maintained on a 12-h light/dark cycle in a temperature-controlled (22°C) barrier facility with free access to

water and a normal diet (NCD, CA-1, CLEA Japan, Inc.). *Alb-CreER^{T2}* mice that express tamoxifen-dependent CreER^{T2} recombinase under the control of the albumin gene promoter were kindly provided by Dr Pierre Chambon (Schuler et al, 2004). We produced *Cnot1^{fl/fl};Alb-CreER^{T2}* mice by crossing *Cnot1^{fl/+};Alb-CreER^{T2}* pairs, which were from mating *Cnot1^{fl/fl}* mice with *Alb-CreER^{T2}* mice. To induce Cre-mediated somatic recombination for deletion of the *Cnot1* gene in adult mice, 6-wk-old *Cnot1^{fl/fl};Alb-CreER^{T2}* mice were fed with a 0.025% tamoxifen-containing normal diet (Research Diets Inc.) for 2 wk, unless otherwise indicated. Tamoxifen-fed *Cnot1^{fl/fl};Alb-CreER^{T2}* mice were used as *Cnot1-LKO* mice. *Alb-CreER^{T2}* and *Cnot1^{fl/fl}* mice were similarly treated as controls. We collected blood samples and measured glucose concentrations with a glucometer (Glutest Pro; Sanwa Kagaku Kenkyusho). Mouse experiments were approved by the Committee of Animal Experiments in the Okinawa Institute of Science and Technology Graduate University and by the Institutional Animal Care and Use Committee at RIKEN, Kobe Branch.

## Antibodies

Mouse monoclonal antibodies against CNOT1, CNOT3, CNOT6L, CNOT8, and CNOT9 were generated by Bio Matrix Research (Suzuki et al, 2015; Takahashi et al, 2015). Antibodies against phospho-JNK (Thr183/Tyr185) (#4671), Cleaved Caspase-3 (Asp175) (#9661), CNOT2 (#34214), RPB1 (#2629), and GAPDH (#2118) were purchased from Cell Signaling Technology. Antibodies against JNK (sc-474), BAX (sc-493), p53 (sc-126), and LAMIN B (sc-6217) were from Santa Cruz Biotechnology. Antibodies against CNOT7 (H00029883-M01A) from Abnova, and RNA Polymerase II (ab817), TBP (ab51841), and IRF9 (ab231015) from Abcam.

## Histological analysis of tissue

After dissection, the livers were fixed with 10% formaldehyde overnight and embedded in paraffin. Sections were stained with Hematoxylin 3G (8656) and Eosin (8659) from Sakura Finetek Japan. Immunohistochemistry for cleaved caspase-3 protein was performed with an antibody against Cleaved Caspase-3 (#9661), as previously described (Suzuki et al, 2019). We captured images and counted the number of apoptotic cells using BZ X-700 (Keyence). We prepared sections from two or three mice of each genotype (*Alb-CreER^{T2}*, *Cnot1^{fl/fl}*, or *Cnot1-LKO*). The representative image of each genotype is shown. H&E–stained sections were analyzed for inflammation, steatosis, and necrosis according to the following scoring system: (i) for inflammation: 0, no inflammation; 1, mild lymphocytic infiltration in the portal triad; 2, severe lymphocytic infiltration in portal triad; 3, extended infiltration of lymphocytes throughout liver; (ii) for steatosis: 0, no steatosis; 1, microsteatosis; 2, microsteatosis and mild macrosteatosis; 3, severe macrosteatosis; (iii) for necrosis: 0, no necrosis; 1, mild necrosis; 2, moderate necrosis; 3, severe necrosis. All scoring was performed by a pathologist blinded to the genotypes.

## Biochemical examination of blood

Plasma for each analysis was obtained by cardiac puncture from deeply anesthetized mice after overnight fasting and was analyzed by the Oriental Yeast Co. Ltd.

## Total RNA-sequencing

Total RNA was isolated from the liver of control (*Alb-CreER^{T2}* and *Cnot1^{fl/fl}*) and *Cnot1-LKO* mice. For comprehensive mRNA half-life profiling, we intraperitoneally injected Act. D (Wako) (2 mg/g body weight) into control (*Cnot1^{fl/fl}*) and *Cnot1-LKO* mice after a tamoxifen-containing diet feeding for 2 wk. We collected livers at 0 h (no injection, n = 4), 4 h (n = 3 in control, n = 4 in *Cnot1-LKO*), and 8 h (n = 3 in control, n = 5 in *Cnot1-LKO*) after injection and extracted total RNAs. At 8 h after Act. D treatment, we did not see any obvious abnormalities in the mice. Total RNA (1 µg) was used for RNA-seq library preparation with a TruSeq Stranded mRNA LT Sample Prep Kit (Illumina), which allows polyA-oligo(dT)–based purification of mRNA, according to the manufacturer's protocol. 109-bp, paired-end read RNA-seq was performed with a HiSeq PE Rapid Cluster Kit v2-HS and a HiSeq Rapid SBS Kit v2-HS (200 Cycle) on a HiSeq 2500 (Illumina), according to the manufacturer's protocol. For data analysis, using StrandNGS software (Strand Genomics, Inc.), reads were mapped to the Ensembl genome sequence (mm10) and FPKMs were calculated. We excluded genes with <0.00015 FPKM. For calculation of mRNA half-lives, *gene* FPKMs were normalized with the FPKM of *Rplp0* mRNA because mRNA quantity per total RNA would decrease upon transcription suppression in Act. D–treated samples. We used the mean of normalized gene FPKMs obtained from independent experiments at each time point, and we calculated mRNA half-lives. The intercept and slope of the linear regression line were applied according to the formula: LN(0.5/e^intercept)/slope (Chen et al, 2008). mRNAs with half-lives less than 0 h or more than 50 h were excluded as unreliable. For calculating intronic FPKMs, reads mapped to the intronic region were extracted with StrandNGS software and read numbers were normalized using the total count of reads mapped to the intronic region and the sum of intron lengths. Genes with <0.01 FPKM and <30-bp intron lengths were eliminated. Sequence data are available through ArrayExpress under the accession number (E-MTAB-5901).

## RIP-sequence (RIP-seq)

Livers from 8-wk-old wild-type mice were homogenized and solubilized in TNE buffer (50 mM Tris–HCl [pH 7.5], 150 mM NaCl, 1 mM EDTA, 1% NP40, and 1 mM PMSF) for 30 min at 4°C. Lysates (150 mg) were incubated with 180 µg of antibodies against CNOT3 and AGO2 (018-22021; Wako) and 100 µl of antibody against BRF1 (#2119; Cell Signaling Technology) for 1 h at 4°C, and then incubated with 1.2 ml of Dynabeads (Invitrogen) for 2 h at 4°C. Total RNAs in immunoprecipitates were isolated using Isogen II. Immunoprecipitates were also analyzed by immunoblotting. Following the manufacturer's protocol, we used 100 ng of total RNA for RNA-seq library preparation with a TruSeq Stranded mRNA Library Prep Kit for NeoPrep (NP-202-1001; Illumina), which allows polyA-oligo(dT)–based purification of mRNA. Minor modification and optimization were

implemented as follows. Custom dual index adaptors were ligated at the 5'- and 3' ends of the library, and PCR was performed for 11 cycles. 150 bp pair-end read RNA-seq was performed with HiSeq 3000/4000 PE Cluster Kit (PE-410-1001; Illumina) and HiSeq 3000/4000 SBS Kit (300 Cycles) (FC-410-1003; Illumina) on HiSeq 4000 (Illumina), according to the manufacturer's protocol. For data analysis, using StrandNGS (Strand Genomics, Inc.), reads were mapped to the Ensembl genome sequence (mm10) and FPKMs were calculated. Genes with <0.1 FPKM in both input and RIP samples were excluded. The RIP enrichment value was calculated by division of gene FPKMs in RIP by that in input total RNA. We used the mean (or median) of the values in three independent expressions, as indicated in figure legends. Sequence data are available through ArrayExpress under accession number (E-MTAB-6941).

## Primary hepatocyte isolation

Adult mice (8-wk-old) were subjected to collagenase perfusion. The liver was perfused with collagenase solution, 18 mM Hepes–NaOH [pH 7.4], 0.075% NaHCO$_3$, 0.5 µg/ml insulin, and 0.1 mg/ml collagenase (C2674; Sigma-Aldrich) in 1× Hank's solution, through the portal vein. Perfused hepatocytes were washed with PBS three times and lysed with TNE buffer.

## Gel filtration chromatography

Liver was lysed with TNE buffer. Lysates (0.5 ml, 3.5 mg/ml) were applied to a Superose 6 10/300 GL column using an AKTA Pure (GE Healthcare). The flow rate was 0.5 ml/min and 0.5 ml fractions were collected.

## Quantitative real-time PCR

Total RNA was isolated from livers using Isogen II (Nippon Gene). cDNA was generated with total RNA (1 µg), oligo (dT) primers (Thermo Fisher Scientific), and SuperScript Reverse Transcriptase III (Thermo Fisher Scientific). We used random primers (Thermo Fisher Scientific) for cDNA synthesis to monitor *Rplp0* and 18S rRNA levels during Act. D treatment (Fig S5). cDNA was mixed with primers and SYBR Green Supermix (Takara) and analyzed with a Viia 7 sequence detection system (Applied Biosystems). Relative mRNA expression was determined after normalization with *Rplp0* or pre-*Rplp0* levels using the ΔΔCt method for mRNAs or pre-mRNAs, respectively. Primers are listed in Table S10.

## ChIP assay

Livers dissected from control (*Cnot1*$^{fl/fl}$) and *Cnot1-LKO* mice were diced into small pieces (~5 mm cubes) and fixed with 1% formaldehyde in phosphate-buffered saline at 37°C for 15 min, followed by two PBS washes. We used anti-RNA Polymerase II antibody and a SimpleChIP Enzymatic Chromatin IP Kit (#9003; Cell Signaling Technology) to prepare nuclear fractions, and for chromatin fragmentation and subsequent immunoprecipitation, with modifications. Briefly, after treatment with 150 units of micrococcal nuclease for 20 min at 37°C, cross-linked chromatin was incubated with 2 µg of anti-RNA Polymerase II antibody (ab817) or ChIP grade

mouse control IgG (ab18143) at 4°C for 24 h. Immunoprecipitates were washed, and DNA was eluted and de–cross-linked as described previously (Takahashi et al, 2012). The final products (DNA fragments) were analyzed by qRT-PCR. We used the same primers as for detection of precursor mRNAs. Primers are listed in Table S10.

## Immunoblotting

Livers were homogenized and solubilized in TNE buffer (50 mM Tris–HCl [pH 7.5], 150 mM NaCl, 1 mM EDTA, 1% NP40, and 1 mM PMSF) for 30 min at 4°C. Lysates dissolved in SDS sample buffer were subjected to SDS–polyacrylamide gel electrophoresis followed by electro-transfer onto Immobilon-P membranes (Millipore). Protein bands were blotted with primary antibodies and ECL anti-rabbit or mouse IgG HRP-linked whole antibody (GE Healthcare) as the secondary antibody. For detection, we used Immobilon Western HRP substrate (Millipore). To quantify the results, we used ImageQuant software in an Image Analyzer LAS 4000 mini (GE Healthcare).

## Subcellular fractionation

Livers were homogenized with hypotonic buffer (10 mM Hepes, 10 mM KCl, and 1.5 mM MgCl$_2$). After centrifugation at 11,000 *g* for 10 min at 4°C, supernatants (cytoplasm) and pellets (nuclei) were dissolved in SDS sample buffer. Nuclear fractions and corresponding amounts of the cytoplasmic fraction were analyzed by immunoblotting. For quantification, ImageJ software was used to measure protein band intensity. Proteins in the nucleus and the cytoplasm were normalized with LAMIN B and GAPDH, respectively.

## Poly(A) tail assay

Poly(A) tail lengths of mRNAs were analyzed using Poly(A) Tail-Length Assay Kits (Affymetrix), according to the manufacturer's protocol. Briefly, 1 µg of total RNA was incubated with poly(A) polymerase in the presence of guanosine (G) and inosine (I) to add a GI tail at the 3'-ends of poly(A)-containing RNAs. cDNA was generated with PAT (PCR poly(A) test) universal primer and reverse transcriptase using GI-tailed RNA as a template. PCR amplification was performed with gene-specific and PAT universal primers and HotStart-IT Taq DNA polymerase. We treated 1 µg of total RNA with 0.2 U of RNase H (Invitrogen), which degrades the RNA strand of RNA-DNA hybrids, at 37°C for 30 min in the presence of 5 µM of the oligo (dT) primer (TTTTTVN; Invitrogen), to remove poly(A) sequences from mRNA and then performed subsequent adaptor ligation, reverse transcription, and PCR reaction. The 0 position of poly(A) tails (A0) was determined from the size of PCR products after RNase H treatment in the presence of oligo (dT). Primers used in these experiments are listed in Table S10. We performed poly(A) tail assays using total RNA prepared from two independent mouse livers in each genotype. For measurements of bulk poly(A) tail lengths, 10 µg of total RNA was labeled with [5'-$^{32}$P] pCp (cytidine 3',5'-bis[phosphate]) (0.11 pmol/µl in total reaction volume 30 µl) (NEG019A; PerkinElmer) using T4 RNA ligase 1 (M0204S; New England Biolabs) at 16°C overnight. Labeled RNAs were incubated at 85°C for 5 min and placed on ice. Then, labeled RNAs were digested with Ribonuclease A (Sigma-Aldrich) and

Ribonuclease T1 (Thermo Fisher Scientific) at 37°C for 120 min in digestion buffer (100 mM Tris–HCl [pH 7.5], 3M NaCl). Reactions were stopped by adding 5× stop solution (10 mg/ml Proteinase K, 0.125 mM EDTA, and 2.5% SDS) and subsequently incubating at 37°C for 30 min. After adding 400 $\mu$l of RNA precipitation buffer (0.5 M NH$_4$OAc and 10 mM EDTA), digested RNA samples were purified by phenol–chloroform extraction and isopropanol precipitation. Final products were fractionated on 8M urea–10% polyacrylamide denaturing gels. Markers (Prestain Marker for small RNA Plus, DM253; BioDynamics Laboratory) were also loaded. Gels were analyzed with a Typhoon FLA 9500 Fluorescence Imager (GE Healthcare). Band intensity was quantified using Image J.

### Bioinformatic analysis

GO enrichment analysis was performed with DAVID Bioinformatics Resources 6.8 (https://david.ncifcrf.gov). GO IDs used in the analyses were transcription (GO:0006351), apoptotic process (GO:0006915), immune system process (GO:0002376), cell cycle (GO:0007049), cellular response to DNA damage stimulus (GO:0006974) and oxidation–reduction process (GO:0055114), and lipid metabolic process (GO:0006629). Violin plots were generated using the R system. Consensus motifs in the promoter region and 3′UTRs were analyzed using HOMER and Amadeus software, respectively. When genes had multiple transcript variants, we first chose transcripts with the smallest Transcript Support Level. When we still had multiple candidates, we chose the one with the longest transcript length.

### Statistical analyses

Comparisons were made using unpaired $t$ test, Wilcoxon rank sum, and Wilcoxon signed-rank tests. Values represent means ± SEM and are represented as error bars. We also used Spearman's correlation coefficient or Pearson's correlation coefficient, as noted in figure legends. Statistical significance is as indicated.

### Accession number

RNA-seq and RIP-seq data are available through ArrayExpress under accession numbers (E-MTAB-5901) and (E-MTAB-6941), respectively.

## Supplementary Information

## Acknowledgements

We thank Dr Pierre Chambon and Dr Daniel Metzger (GIE-CERBM) for providing *Alb-CreER*$^{T2}$ mice. We thank Okinawa Institute of Science and Technology Graduate University (OIST) for generous support to the Cell Signal Unit. We also thank the OIST DNA Sequencing Section for preparing Illumina sequence libraries and for performing HiSeq sequencing. This work was supported by a Grant-in-Aid for Scientific Research (S) (21229006), Scientific Research (C) (17K07292, 18K06975, 18K07079), Japan Society for the Promotion of Science Fellows (10J08349), Young Scientists (B) (25860761), and Scientific Research on Innovative Areas in a proposed research area (25121734, 17H06018) from the Japan Ministry of Education, Culture, Sports, Science and Technology. This work was also supported by a Grant from Joint Research Project of the Institute of Medical Science, the University of Tokyo.

## Author Contributions

A Takahashi: conceptualization, validation, and investigation.
T Suzuki: formal analysis, validation, and investigation.
S Soeda: data curation, validation, and investigation.
S Takaoka: data curation, formal analysis, and validation.
S Kobori: formal analysis and validation.
T Yamaguchi: investigation.
HMA Mohamed: validation and investigation.
A Yanagiya: formal analysis and investigation.
T Abe: resources.
M Shigeta: resources.
Y Furuta: resources.
K Kuba: data curation.
T Yamamoto: conceptualization.

## Conflict of Interest Statement

The authors declare that they have no conflict of interest.

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
