## [Reviewer comments · Life Science Alliance]

Life Science Alliance

The CCR4-NOT complex maintains liver homeostasis through mRNA deadenylation

Akinori Takahashi, Toru Suzuki, Shou Soeda, Shohei Takaoka, Shungo Kobori, Tomokazu Yamaguchi, Haytham Mohamed, Akiko Yanagiya, Takaya Abe, Mayo Shigeta, Yasuhide Furuta, Keiji Kuba, and Tadashi Yamamoto

DOI: <https://doi.org/10.26508/lsa.201900494>

Corresponding author(s): Tadashi Yamamoto, Okinawa Institute of Science and Technology Graduate University and Toru Suzuki, Center for Integrative Medical Sciences, RIKEN

Review Timeline:

Submission Date:	2019-07-19
Editorial Decision:	2019-08-14
Revision Received:	2020-02-13
Editorial Decision:	2020-03-03
Revision Received:	2020-03-11
Accepted:	2020-03-14

Scientific Editor: Andrea Leibfried

Transaction Report:

August 14, 2019

Re: Life Science Alliance manuscript #LSA-2019-00494-T

Dr. Tadashi Yamamoto
Okinawa Institute of Science and Technology Graduate University
Cell Signal Unit
1919-1 Tancha
Okinawa 904-0495
Japan

Dear Dr. Yamamoto,

Thank you for submitting your manuscript entitled "The CCR4-NOT complex maintains liver homeostasis through mRNA deadenylation" to Life Science Alliance. The manuscript was assessed by expert reviewers, whose comments are appended to this letter.

As you will see, all three reviewers appreciate your analyses. Reviewer #1 and #2, however, point out that your conclusions are currently not sufficiently supported by the data provided. All reviewers provide constructive input on how to strengthen your work, in large parts by changing the presentation of your findings and by adding more information/clarifications. We would thus like to invite you to submit a revised version of your manuscript to us. Importantly, reviewer #1 notes that there are issues with your conclusion regarding transcription rates. Please follow the reviewer's suggestion on how to address this issue. Reviewer #2 thinks that your data may be compromised by contaminating immune cell infiltrations. Again, the reviewer guides you how to address this issue via toning down your statements and adding information on immune cell infiltrations. The requested control should get added as well, while the co-IP (for Fig 5I) requested by reviewer #2 is not mandatory for acceptance here.

Thank you for this interesting contribution to Life Science Alliance. We are looking forward to receiving your revised manuscript.

Sincerely,

B. MANUSCRIPT ORGANIZATION AND FORMATTING:

Reviewer #1 (Comments to the Authors (Required)):

In this manuscript Takahashi and colleagues analyze the impact of Cnot1 inactivation in mice. The deletion of Cnot1 being lethal at an early embryonic stage, the authors focused on the liver by inducing Cnot1 deletion in a tissue-specific manner. This treatment induced lethal hepatitis demonstrating an essential function for Cnot1 in the mouse liver. Cnot1 depletion resulted in cell death with concomitant accumulation of mRNAs with longer poly(A) tails. Transcriptome analyses indicated that mRNAs encoding transcription factors as well as proteins involved in apoptosis and immune functions accumulated while mRNAs for liver specific functions were relatively underrepresented. mRNAs from the former group were also co-precipitated with Cnot3. For some of those, interactions were likely to involve Ago2 or Brf1. The authors observed that the CCR4-NOT complex was cytoplasmic in mouse liver cells but noticed changes in the levels of pre-mRNA sequences in the cellular transcriptome after Cnot1 depletion and interpreted the latter as an indirect impact on transcriptional rates.

Overall, this manuscript presents interesting new data. If this analysis provides overall only limited new mechanistic insights into the mechanism of action of the CCR4-NOT complex, it clearly demonstrates that the Cnot1 protein is essential for mouse development and liver function. Consistent with the reported deadenylation activities of 2 CCR4-NOT subunits and the cytoplasmic localization of this complex observed by the authors, the absence of Cnot1 leads to changes in mRNA turnover accompanied by accumulation of mRNAs with longer poly(A) tails. Altogether, the authors present solid data demonstrating that alteration of deadenylation resulting from altered Cnot1 level result in liver dysfunction leading to lethal hepatitis.

If this study is generally sound, some interpretation made by the authors should be taken with caution. Hence, authors use the abundance of pre-mRNA sequences as a proxy for transcription rates. However, changes in pre-mRNA levels following Cnot1 depletion may not only result from changes in transcription but could also be affected indirectly by changes in the levels of general or specific splicing factors, modulation of nuclear RNA decay and/or of nuclear RNA export. It is thus not possible to definitively interpret the data shown as demonstrating that transcription was altered. Authors should more carefully present their data and interpretation.

Similarly, the authors use Rplp0, Rps3a1 or Gapdh to normalize transcriptome analyses. Besides assuming that these transcripts are unaffected by Cnot1 depletion, the authors make the hypothesis that the mRNA quantity per cell is identical in wild-type and Cnot1-depleted liver. This may not be the case! If so this would question the conclusions presented. Hence, the decreased level of mRNAs encoding liver specific functions may only be relative to other mRNAs with the absolute level of all mRNAs increasing.

Altogether, I believe that this manuscript present interesting observations that will be of rather wide interest. However, the manuscript needs to be revised in depth before a possible publication to carefully present the data and their interpretation taking into accounts comments above and below.

Detailed comments:

- English should be checked.
- The authors analysis demonstrate again that Cnot1 is required for deadenylation in mammalian cells. This was not necessarily expected because the two catalytic subunits of the CCR4-NOT complex do not require Cnot1 for poly(A) tail degradation. Authors should probably discuss this point that parallels findings reported in yeast.

- Abstract: " Many of mRNAs encoding metabolic enzymes become less abundant upon Cnot1 suppression due largely to transcriptional downregulation" is not demonstrated by the experiments presented (see above).
- Throughout the manuscript: Authors should be more cautious about their statements. For example, in the introduction they indicate "Mechanistically, the CCR4-NOT complex targets TF-, apoptosis- and immune-mRNAs through its association with Ago2 and Brf1 in liver" Should read "Mechanistically, the CCR4-NOT complex targets TF-, apoptosis- and immune-mRNAs MAINLY through its association with Ago2 and Brf1 in liver" as additional factors may be involved. This should be checked throughout the text.
- Introduction, page 3, line 59: More recent reviews have been published e.g. by Gross and colleagues.
- Introduction, page 3, line 60: As presented by the authors, deadenylation is principally followed by exosome degradation rather than 5'-3' decay. The opposite is believed to be the main pathway.
- Introduction, page 4, line 74: evidences that Cnot1 and homologs act as a scaffold have been substantiated by different structural studies in different species that should probably be all quoted.
- Introduction, page 5, line 106: "mainly due to reduced transcription" is exaggerated.
- Results, page 6, line 121: Authors should comment on the number of individuals at the different stages presented in Supplementary Figure 1E. Is a fraction of the heterozygous embryos also dying early?
- Results, page 6, line 123: Unclear because the cross indicated will not generate a Cnot1 KO in the liver.
- Results, page 7 and 8, line 154-158: Knowing the amount of RNA per cell is necessary to conclude "upregulated" or "downregulated". Moreover, given that no regulation is shown, authors should better use "accumulated", "underrepresented" or equivalent.
- Results, page 9, line 184: "regulated" should be changed as no regulation is shown (regulation entails a signal inducing a change). "controlled" would be more appropriate.
- Results, page 10, line 204: "regulated" should be changed.
- Results, page 11, line 234: "Apparently, CCR4-NOT complex requires RNA-binding proteins" is meaningless.
- Results, page 11, line 242: U- or A- rich motifs should be defined.
- Results, page 12, line 246 and 248: There is no Figure 5J.
- Results, page 12, line 249 and figure 5I: The fate of numerous transcripts is not explained by Ago2 or Brf1. This suggests implication of additional factors. This should be mentioned and discussed.
- Results, page 12, line 263 and supplementary figure 5: This refers to a log/log fit that are not very accurate. What is the correlation if these data are analyzed using linear scales?
- Discussion, page 17, line 364: The proposed increase in total transcriptional activity may simply reflect reduced splicing.

Reviewer #2 (Comments to the Authors (Required)):

The work by Takahashi et al. describes CCR4-NOT deadenylase relevance in liver homeostasis. The authors characterize how liver-specific disruption of Cnot1 in adult mice leads to an elongation of bulk RNA poly(A) tail length, and a global increase in both steady state RNA levels and RNA stability. The relevance of Cnot1 contribution to the regulation of gene expression is demonstrated by the development of lethal liver damage in Cnot1LKO mice. The authors report that, upon hepatocyte-specific Cnot1 depletion, TF-, apoptosis- and immune-mRNAs are upregulated in total liver extract due to their increased transcription and mRNA stability; while metabolic-mRNAs are downregulated due to a transcriptional downregulation. The authors propose that the mechanisms by which the CCR4-NOT regulates TF-, apoptosis- and immune-mRNAs is through its association

with Ago2 and Brf1.

The characterization of a liver-specific disruption of Cnot1 is clearly relevant, and the following genome wide analyses would be of interest to the field. However the work, in its current format, has relevant limitations. As detailed below some could be solved with more experimental details and others with deeper analyses. However, in my opinion, there is a major limitation on the strategy of the study arising from the genomic studies in total liver, when there is clearly more immune infiltration in the hepatocyte specific KO. Thus, some of the presented observations could be derived from changes in the cellular populations in Cnot1KO livers. That makes very difficult to distinguish direct (hepatocyte) effects from indirect (Niche) effects. From that point on, the model proposed is not fully supported by the data presented. Moreover, the relevance of Ago2 and Brf1 in the Cnot1KO phenotype or the idea that metabolic mRNAs are not strictly regulated by the CCR4-NOT in the liver are not unequivocally supported by the data presented. Finally, I have also missed more details about the quality of the genome-wide experiments and its replicates, specifically more statistical analysis, quality controls and a more detailed explanation of how the analysis has been performed.

In my opinion, presenting the data in a different form and limiting the scope of the conclusions presented would result in a stronger work. Alternatively, the genome wide analyses should be done in isolated hepatocytes, not total liver.

Major concerns:

- Figure 1. For a better interpretation of the rest of results, it would be worth to better characterize the phenotypes (steatosis, fibrosis, transaminases, immune-cell staining) the authors observe in Cnot1KO livers upon tamoxifen diet feeding.

The authors should quantify the differences in immune infiltration and cell death they observe. Defining the observed liver damage as Hepatitis will require a much deeper characterization of the phenotype.

The abstract might misled the reader into thinking that lethal hepatitis is caused by the upregulated mRNAs, but causality is not demonstrated.

It would be worth to frame the obtained results to the specific timepoint at which the experiments were done (14 days after tamoxifen addition, when 100% of the animals do not survive longer than 17 days according to figure1C). Therefore, the GO categories in which the authors focused Cnot1 hepatic function could be specific of a damaged liver (dying animals) and not necessarily reflect a direct Cnot1 function in homeostatic livers.

In figure 1l the loading control is missing.

- Figure 3A. The authors should provide clear information on how the analysis of replicates was performed and how significant were the changes between the transcriptome of WT and Cnot1LKO livers (even better in isolated hepatocytes). It would be worth to perform this analysis with the standard methods Deseq2, EdgeR or Limma-Voom and show in figures like Fig3A: also, please indicate which genes are changing significantly.

Related to the analysis, the authors show in Figure 2B how Gapdh poly(A) tail is longer in Cnot1LKO. Could the authors show that Rplp0, Gapdh and Rps3a1 are appropriate genes for normalization? As an alternative, the previously proposed methods (Deseq2...) do not use a single mRNA for normalization.

The authors attribute the enrichment in apoptosis- and immune-mRNAs to the lack of Cnot1-

mediated destabilization of these mRNAs. To validate this hypothesis, they should test the stability or poly(A) tail of these mRNAs specifically in hepatocytes in order to rule out that the enrichment on these mRNAs is derived from the differences in apoptotic and immune cells between wild type and Cnot1KO livers.

- Figure 4A. In the methods section, could the authors specify how did they analyse the different replicates? In Fig4A and Fig4C could they also specify the number of mRNAs that are considered and further explain how did they perform the ranking of genes for the X axis.

The authors claim that metabolic genes are not strictly regulated by the CCR4-NOT complex because of their lower enrichment in the RIP/Input ratio. It may be worth to reconsider this interpretation as several of the presented results suggest that they are indeed regulated by Cnot1 (Fig2B, Fig5C).

Additionally, I would suggest that normalization of the RIP should be done as enrichment over IgG or over Cnot1-KO not over the input. As done, the abundant RNAs will show a low RIP enrichment score. Because metabolic mRNAs are more abundant than the other GO categories studied (Fig3C), their RIP value is underestimated. Lower IP/Input ratio could be caused by differences on Input FPKM.

The authors should clearly acknowledge that not necessarily all CCR4NOT RIP targets are from hepatocytes. Indeed, a RIP in isolated hepatocytes would have generated more interpretable results. It would be useful to estimate/comment which percentage of the immunoprecipitated CCR4-NOT complex comes from hepatocytes vs. other cell populations.

- Figure 5A. In the genome wide assessment of mRNA half-lives there is no information about how the replicates were treated or the significance of the changes described. I would also encourage the authors to show the decay of specific mRNAs in order to proof that the Actinomycin D pulse-chase assay worked properly and/or compare the obtained half-lives with previous data in the literature.

- Figure 5I. The authors claim that CCR4-NOT targets TF-, apoptosis-, and immune-mRNAs through its association with Ago2 and Brf1. The only result supporting this statement is the overlap in RIP targets, which does not show direct association between CNOT and these proteins. The authors could perform co-IPs in the presence or absence of RNA to demonstrate this interaction. Target overlapping is consistent but not a proof of mechanism as implicated. Additionally, the authors should show the immunoblots of the Ago2 and Brf1 IP and validate that their obtained list of targets is specific by showing that previously defined targets of these proteins are enriched, while negative controls are not. Finally, they should consider reformulating this statement as it does not rule out that other RBPs are being the ones directing CCR4-NOT to these mRNAs.

- Figure 6B. In the genome-wide assessment of intron reads, the authors should provide information about how the replicates were treated or the statistical significance of the observed alterations. According to the materials and methods, no normalization by the library size was applied, which could have led to a misinterpretation of the obtained results.

- Figure 6C. The authors propose that total transcriptional activity is increased in Cnot1KO livers. Given the minor difference that is shown and the fact that there are mRNAs showing opposite behaviours (Figure 6C-G), further evidence should be provided in order to make this statement.

Minor comments:

- Figure 2B. Given that the authors claim that most of Cnot1 effects are in immune-related, TF- or apoptosis- mRNAs, it would be an added value to demonstrate that these mRNAs present alterations in their poly(A) length.
- Figure 3C. Is the downregulation in highly expressed metabolic mRNAs specific of metabolic mRNAs? Are highly expressed immune- or apoptosis- mRNAs also downregulated?
- The results section citing figure 5G-I should be revised (text and figures do not match).

Reviewer #3 (Comments to the Authors (Required)):

The authors use inducible CNOT1 KO mice and global analyses to demonstrate a role for the deadenylase complex in liver biology. The data are generally convincing (e.g. the changes in poly(A) tail length in Fig. 2 are particularly well done) and support the conclusions drawn. I only request a couple of clarifications to polish the presentation and put it in full context of the field:

Major Points:

1. Fig. 4: More details need to be presented as to how the mRNA half life experiment was done. For example, what time points after actD administration were analyzed? What effect did the actD treatment have on the mice during the course of the experiment? ActD is known, of course, to generate major changes in cellular physiology - thus it is very important for the details of this experiment to be presented so that the data can be put into context.
2. The authors note that there is not a strict association between changes in mRNA half life and changes in mRNA abundance. While they descriptively report on changes in transcription rates, they do not entertain the possibility that this may be due to a 'transcriptional buffering' phenomenon in which the synthetic rate of a transcript is altered by the cell to adjust expression when mRNA decay rates change. If this is indeed the case, it is probably worth at least a paragraph in the discussion to document this buffering phenomenon/mechanism as it is relatively underreported in mammalian systems to date.

Minor Points:

1. Fig 2A/Results section/line 142-143: In the sentence that starts. " In Cnot1-KO livers, the population of poly(A) tail length with longer than 70 nt dramatically increased' , I don't really understand what the authors mean by the next phrase, "instead of decrease in that with shorter than 70 nt ." I would recommend rewriting that last part.
2. Fig 4: I'm a bit confused as to what the x axes indicate on the graphs in panels A and C. Please clarify in the legend for the reader.

Reviewer 1

However, changes in pre-mRNA levels following Cnot1 depletion may not only result from changes in transcription but could also be affected indirectly by changes in the levels of general or specific splicing factors, modulation of nuclear RNA decay and/or of nuclear RNA export. It is thus not possible to definitively interpret the data shown as demonstrating that transcription was altered. Authors should more carefully present their data and interpretation.

Similarly, the authors use Rplp0, Rps3a1 or Gapdh to normalize transcriptome analyses. Besides assuming that these transcripts are unaffected by Cnot1 depletion, the authors make the hypothesis that the mRNA quantity per cell is identical in wild-type and Cnot1-depleted liver. This may not be the case! If so this would question the conclusions presented. Hence, the decreased level of mRNAs encoding liver specific functions may only be relative to other mRNAs with the absolute level of all mRNAs increasing.

1. We agree that we cannot exclude possible involvement of other mechanisms, such as splicing, nuclear RNA decay, or nuclear RNA export in the changes in pre-mRNA levels following CNOT1 suppression, though previous reports showed that changes in pre-mRNA levels are significantly correlated with transcription rate (Gaidatzis et al, doi:10.1038/nbt.3269; Wang et al, doi/10.1073/pnas.1715225115). We have now carefully considered this point and have discussed it relative to our results. Basically, we have stated that there was an increase of pre-mRNA levels and we have discussed several possible reasons (page 20, line 417-428) in the revised manuscript.

2. We agree with the reviewer and we rewrote the manuscript as follows. It is also possible that decreased levels of mRNAs encoding liver function-related molecules are only relative to other mRNA species that increased significantly in livers of *Cnot1-LKO* mice. This is described in page 20, line 424-428 of the revised manuscript. In addition, we dropped the normalization of RNA-seq data against a specific mRNA, as also suggested by Reviewer 2 (See Figure 3A in the revised manuscript).

- English should be checked.

We employed a professional technical editor to correct the English.

- The authors analysis demonstrate again that Cnot1 is required for deadenylation in mammalian cells. This was not necessarily expected because the two catalytic subunits of the CCR4-NOT complex do not require Cnot1 for poly(A) tail degradation. Authors should probably discuss this point that parallels findings reported in yeast.

Of course, the catalytic subunits exhibit deadenylase activity as a single molecule *in vitro*. On the other hand, previous studies and our present findings about the functions of CNOT1 in both yeast genetic analyses and mammalian systems strongly suggest that CNOT1 is essential to mRNA deadenylation "*in vivo*." We have proposed only the essential *in vivo* roles (page 7, line 116-118 and page 18, line 362-367) in the revised manuscript.

- Abstract: " Many of mRNAs encoding metabolic enzymes become less abundant upon

Cnot1 suppression due largely to transcriptional downregulation" is not demonstrated by the experiments presented (see above).

We have changed the explanation, saying that the results are “concomitant with a decrease of their immature, unspliced mRNAs” (page 3, line 38-41) in the revised manuscript.

- Throughout the manuscript: Authors should be more cautious about their statements. For example, in the introduction they indicate "Mechanistically, the CCR4-NOT complex targets TF-, apoptosis- and immune-mRNAs through its association with Ago2 and Brf1 in liver" Should read "Mechanistically, the CCR4-NOT complex targets TF-, apoptosis- and immune-mRNAs MAINLY through its association with Ago2 and Brf1 in liver" as additional factors may be involved. This should be checked throughout the text.

We have moderated such statements throughout the text, and changed the description to explain our results more tentatively.

- Introduction, page 3, line 59: More recent reviews have been published e.g. by Gross and colleagues.

We have added “Mugridge et al., Nat. Struct. Mol. Biol. 25, 1077-1085. doi: 10.1038/s41594-018-0164-z” (page 4, line 64) in the revised manuscript.

- Introduction, page 3, line 60: As presented by the authors, deadenylation is principally followed by exosome degradation rather than 5'-3' decay. The opposite is believed to be the main pathway.

We have revised the sentence (page 4, line 65-67) in the revised manuscript.

- Introduction, page 4, line 74: evidences that Cnot1 and homologs act as a scaffold have been substantiated by different structural studies in different species that should probably be all quoted.

We have added 3 references on page 5 lines 75 and 79-81 in the revised manuscript.

- Introduction, page 5, line 106: "mainly due to reduced transcription" is exaggerated.

We have changed the description here (page 6, line 107-110) in the revised manuscript.

- Results, page 6, line 121: Authors should comment on the number of individuals at the different stages presented in Supplementary Figure 1E. Is a fraction of the heterozygous embryos also dying early?

In the revised manuscript, we have stated that, after crossing *Cnot1*^{+/-} pairs, wild-type and *Cnot1*^{+/-} mice were born in approximately a 1:1 ratio, and normally grew to adulthood, while *Cnot1*^{-/-} mice die in embryo (page 7, line 124-126).

- Results, page 6, line 123: Unclear because the cross indicated will not generate a Cnot1 KO in the liver.

We rewrote the procedure (page 7, line 129-133) in the revised manuscript. The details are in the Materials and Methods.

- Results, page 7 and 8, line 154-158: Knowing the amount of RNA per cell is necessary

to conclude "upregulated" or "downregulated". Moreover, given that no regulation is shown, authors should better use "accumulated", "underrepresented" or equivalent.

We have now used "increased" and "decreased" consistently throughout the manuscript.

- Results, page 9, line 184: "regulated" should be changed as no regulation is shown (regulation entails a signal inducing a change). "controlled" would be more appropriate. Following this suggestion, we have replaced "regulated" with "controlled" (page 10, line 202) in the revised manuscript.

- Results, page 10, line 204: "regulated" should be changed.

The paragraph in question, including this sentence, was largely revised. Please see page 10 line 198 to page 12, line 231, in the revised manuscript.

- Results, page 11, line 234: "Apparently, CCR4-NOT complex requires RNA-binding proteins" is meaningless.

We deleted the sentence.

- Results, page 11, line 242: U- or A- rich motifs should be defined.

This is defined in Supplementary Figure 6 in the revised manuscript. That figure was included as Supplementary Figure 4B in the original manuscript. We described the motifs in the text (page 14, line 292-294)

- Results, page 12, line 246 and 248: There is no Figure 5J.

We are glad that the reviewer caught this mistake. Figures 5 and 6 are restructured and renumbered in the revised manuscript.

- Results, page 12, line 249 and figure 5I: The fate of numerous transcripts is not explained by Ago2 or Brf1. This suggests implication of additional factors. This should be mentioned and discussed.

We agree, and we rewrote the manuscript by discussing possible involvement of other RNA binding proteins (page 21, line 437-443) in the revised manuscript. Please also see our response to Reviewer 2 (line 333-335 in this text).

- Results, page 12, line 263 and supplementary figure 5: This refers to a log/log fit that are not very accurate. What is the correlation if these data are analyzed using linear scales?

There was a very wide range of values in the measured and calculated data, leading us to use log/log plotting. Because read count data in RNA-seq do not fit a normal distribution, we used a non-parametric method (Spearman's rank correlation coefficient) to measure the strength of association between FPKM and (half-life x intron FPKM). Even when we assume a normal distribution and measure the strength of linear correlation, there is weak, but significant positive correlation between them ($\rho = 0.301$, 0.286 in control and KO data, respectively, in Pearson's correlation coefficient, $P < 0.001$). These data appear in Supplementary Figure 7A in the revised manuscript.

- Discussion, page 17, line 364: The proposed increase in total transcriptional activity may simply reflect reduced splicing.

Following this suggestion, the effect of splicing is also considered in the revised manuscript. Please see page 20, line 417-428 in the revised manuscript.

Reviewer 2

- Figure 1. For a better interpretation of the rest of results, it would be worth to better characterize the phenotypes (steatosis, fibrosis, transaminases, immune-cell staining) the authors observe in *Cnot1*KO livers upon tamoxifen diet feeding.

The authors should quantify the differences in immune infiltration and cell death they observe. Defining the observed liver damage as Hepatitis will require a much deeper characterization of the phenotype.

To characterize the phenotype of *Cnot1-LKO* liver, we have analyzed inflammation, steatosis, and necrosis, using HE-stained sections, according to the scoring system described in the Materials and Methods. The analyses were performed by a pathologist. The results show that inflammation and hepatic necrosis were very severe in livers from *Cnot1-LKO* mice (Table 1). Steatosis and fibrosis were not prominent in all samples. We have also calculated the number of apoptotic hepatocytes in *Cnot1-LKO* mice using cleaved caspase-3-stained liver sections (Figure 1I). In addition, we performed biochemical analyses of blood, using serum from control and *Cnot1-LKO* mice. AST, ALT, ALP and LDH were all significantly increased in *Cnot1-LKO* mice (Table 2). The description is given on page 8, line 140-151 and in the Materials and Methods in the revised manuscript.

It would be worth to frame the obtained results to the specific timepoint at which the experiments were done (14 days after tamoxifen addition, when 100% of the animals do not survive longer than 17 days according to figure 1C). Therefore, the GO categories in which the authors focused *Cnot1* hepatic function could be specific of a damaged liver (dying animals) and not necessarily reflect a direct *Cnot1* function in homeostatic livers.

According to the reviewer's comment, we have added a description about the possibility that the GO categories enriched in *Cnot1-LKO* mice could be specific for damaged liver (dying animals) and may not necessarily reflect a direct *Cnot1* function in homeostatic livers. These matters are explained on page 10, line 198-201 in the revised manuscript.

In figure 1I the loading control is missing.

We used the same liver lysates for both Figures 1A and 1I. To avoid misunderstanding, we have combined them into one figure (Figure 1A) in the revised manuscript.

- Figure 3A. The authors should provide clear information on how the analysis of replicates was performed and how significant were the changes between the transcriptome of WT and *Cnot1*LKO livers (even better in isolated hepatocytes). It would be worth to perform this analysis with the standard methods Deseq2, EdgeR or Limma-Voom and show in figures like Fig3A: also, please indicate which genes are changing significantly.

Related to the analysis, the authors show in Figure 2B how *Gapdh* poly(A) tail is longer in *Cnot1*LKO. Could the authors show that *Rplp0*, *Gapdh* and *Rps3a1* are appropriate

genes for normalization? As an alternative, the previously proposed methods (Deseq2...) do not use a single mRNA for normalization.

Instead of normalization of RNA-seq data with a single mRNA, we have analyzed our data using FPKM for normalization, which is one of the standard methods for RNA-seq. Four biological replicates (in both control and *Cnot1-LKO* mice) were used, and the averages among the four samples are shown in Figure 3A in the revised manuscript. To identify differently expressing genes (DEG) between control and *Cnot1-LKO* mice, we used the Mann-Whitney U-test. We defined genes showing more than 2-fold difference in expression with FDR <0.05 as DEGs (red and blue dots). The analyses show that mRNAs involved in “transcription”, “cell cycle” and “response to DNA damage stimulus” were prominently enriched GO terms among the increased mRNAs in *Cnot1-LKO* mice. GO terms “apoptosis” and “immune system process” were significantly enriched, but were not among the top five GO terms. Therefore, we have changed the description throughout the text.

The authors attribute the enrichment in apoptosis- and immune-mRNAs to the lack of *Cnot1*-mediated destabilization of these mRNAs. To validate this hypothesis, they should test the stability or poly(A) tail of these mRNAs specifically in hepatocytes in order to rule out that the enrichment on these mRNAs is derived from the differences in apoptotic and immune cells between wild type and *Cnot1KO* livers.

This is very important comment. However, due to technical limitations in hepatocyte isolation using livers from *Cnot1-LKO* mice, we could not perform the suggested experiments. It was extremely hard to prepare hepatocytes from hepatitis livers.

Please note that our focus shifted from apoptosis-mRNAs and immune-mRNAs to mRNAs encoding TFs, cell cycle regulators, and DNA damage response-related molecules, according to the results of GO analysis using a different normalization method, as suggested by the reviewer (please see line 185-196 in this text). Results of RIP-qPCR using isolated, wild-type hepatocyte lysates showed that the CCR4-NOT complex bound to mRNAs for TFs, cell cycle regulators, and DNA damage response-related molecules in isolated hepatocytes, as well as in whole liver (Supplementary Figure 3C, D), suggesting that poly(A) tail elongation and stabilization of those mRNAs occurred in hepatocytes (page 11, line 214-219 of the revised manuscript). On the other hand, binding of the CCR4-NOT complex to “immune system process”-related mRNAs (*Cxcl10* and *Tlr3*) was not significant in isolated hepatocytes (Supplementary Figure 3D). On the basis of these results, together with the polyA tail analysis of *Cxcl10* mRNA, pre-mRNA expression, and ChIP experiments (Supplementary Figure 2B and Figure 7 B, D), we attributed an increase of at least some “immune system process”-related mRNAs to transcriptional activation (page 16, line 334-335 of the revised manuscript). We also noted the possibility that the increase is derived from an infiltration of immune cells in livers of *Cnot1-LKO* mice (page 11, line 221-223, and from page 15, line 318 to page 16, line 334 of the revised manuscript).

- Figure 4A. In the methods section, could the authors specify how did they analyse the different replicates? In Fig4A and Fig4C could they also specify the number of mRNAs that are considered and further explain how did they perform the ranking of genes for

the X axis. The authors claim that metabolic genes are not strictly regulated by the CCR4-NOT complex because of their lower enrichment in the RIP/Input ratio. It may be worth to reconsider this interpretation as several of the presented results suggest that they are indeed regulated by Cnot1 (Fig2B, Fig5C).

Additionally, I would suggest that normalization of the RIP should be done as enrichment over IgG or over Cnot1-KO not over the input. As done, the abundant RNAs will show a low RIP enrichment score. Because metabolic mRNAs are more abundant than the other GO categories studied (Fig3C), their RIP value is underestimated. Lower IP/Input ratio could be caused by differences on Input FPKM.

1. We normalized gene FPKM in RIP against input total RNA (defined as the CCR4-NOT RIP enrichment value), and calculated mean CCR4-NOT RIP enrichment values from three independent experiments. mRNAs were ordered according to their CCR4-NOT RIP enrichment values. In Figure 4A, the X axis represents ranking in ascending order. Similar results were obtained when we used the medians of both CCR4-NOT RIP enrichment value and RIP FPKM (Supplementary Figure 4).
2. We added the number of mRNAs analyzed to Figure 4A, C.
3. For Figure 4C (Figure 4F in the revised manuscript), the way of calculating mRNA half-lives is described in the Materials and Methods. mRNAs are ordered according to their half-lives (h). The X axis represents ranking in ascending order. Please see also our responses about Figure 5A (line 285-298 in this text) and to Reviewer 3, comment 1.
4. We think that FPKMs in control IgG RIP represent only non-specific binding of mRNAs and should not be used for normalization. While normalization of the CCR4-NOT RIP in control over *Cnot1-LKO* is much better way, we would like to understand “the relation between expression level and binding of the CCR4-NOT complex” for each mRNA. In the revision, we carefully changed the description on page 10, line 198 to page 12, line 231 in the revised manuscript, so as not to mislead readers.

As the reviewer observed, our interpretation of the RIP-seq results of metabolic genes was not correct. We acknowledge that the results of RIP experiments showed that metabolic genes were recognized by the CCR4-NOT complex, albeit with low efficiency. Moreover, mRNA half-life data obtained from Act. D chase experiments showed that metabolic mRNAs were obviously stabilized in livers from *Cnot1-LKO* mice. We have clearly stated that metabolic and other genes are controlled by the CCR4-NOT complex in liver (page 13, line 262-266) in the revised manuscript.

The authors should clearly acknowledge that not necessarily all CCR4NOT RIP targets are from hepatocytes. Indeed, a RIP in isolated hepatocytes would have generated more interpretable results. It would be useful to estimate/comment which percentage of the immunoprecipitated CCR4-NOT complex comes from hepatocytes vs. other cell populations.

We have performed qPCR using CCR4-NOT RIP products prepared from lysates of isolated wild-type hepatocytes. The results showed that the CCR4-NOT complex bound to mRNAs encoding TFs, cell cycle regulators, and DNA damage response-related

molecules in isolated hepatocytes, as well as whole liver (Supplementary Figure 3C, D). On the other hand, binding of the CCR4-NOT complex to “immune system process”-related mRNAs (*Cxcl10* and *Tlr3*) was not significant in isolated hepatocytes (Supplementary Figure 3D). Therefore, it is possible that several mRNAs detected in CCR4-NOT RIP using whole liver, in particular “immune system process”-related mRNAs, are from cells other than hepatocytes. We have pointed this matter on page 11, line 214-223 in the revised manuscript.

- Figure 5A. In the genome wide assessment of mRNA half-lives there is no information about how the replicates were treated or the significance of the changes described. I would also encourage the authors to show the decay of specific mRNAs in order to proof that the Actinomycin D pulse-chase assay worked properly and/or compare the obtained half-lives with previous data in the literature.

1. In actinomycin D (Act. D) chase experiments, we used gene expression values (gene FPKM normalized with *Rplp0* FPKM) at each time point (0, 4, 8h) to calculate half-lives. We believe that normalization using FPKMs is not meaningful in this analysis, because total mRNA quantity per cell would decrease upon transcription suppression (Figure 4C). We collected livers at 0h (n=4), 4h (n=3 in control, n=4 in *Cnot1-LKO*) and 8 h (n=3 in control, n=5 in *Cnot1-LKO*) after Act. D injection and extracted total RNAs. Different mice were used at each time point. In these sets of experiments, we cannot determine mRNA half-lives in each experiment; thus, we calculated mean gene expression values from the independent experiments at each time point and used the means for calculating mRNA half-lives (please see the Materials and Methods). We simply compared the calculated half-lives between control and *Cnot1-LKO* mice (Figure 5A). We performed the same experiments and analyses in Figures 4C and 5A. Please see also our response to the comment about Figure 4C (line 245-249 in this text) and to Reviewer 3’s comment #1.
2. We have compared the half-lives in our study with previously reported data using chase experiments (Sharova et al. 2009; Friedel et al. 2009; Schwanhausser et al, 2011). Although the data are from NIH3T3 and mouse embryonic stem cells, our results were weakly, but significantly correlated with them (Supplementary Figure 5A). These data suggest that treatment of mice with Act. D treatment worked. These results are described on page 12, line 235-241 in the revised manuscript. In addition, we plotted decay curves of several mRNAs using their read counts in RNA-seq data (Supplementary Figure 5B).

- Figure 5I. The authors claim that CCR4-NOT targets TF-, apoptosis-, and immune-mRNAs through its association with Ago2 and Brf1. The only result supporting this statement is the overlap in RIP targets, which does not show direct association between CNOT and these proteins. The authors could perform co-IPs in the presence or absence of RNA to demonstrate this interaction. Target overlapping is consistent but not a proof of mechanism as implicated.

The interaction of the CCR4-NOT complex with AGO2 or BRF1 already been shown in the published papers we cited in the manuscript. We agree with the comment that target overlapping is not a proof of mechanism, and we rewrote the manuscript.

Additionally, the authors should show the immunoblots of the Ago2 and Brf1 IP and validate that their obtained list of targets is specific by showing that previously defined targets of these proteins are enriched, while negative controls are not.

Finally, they should consider reformulating this statement as it does not rule out that other RBPs are being the ones directing CCR4-NOT to these mRNAs.

We performed immunoblots of AGO2 and BRF1 IP (Figure 6A). We analyzed BRF1 RIP-RNA-seq data and found that 25 mRNAs among the top 30 enriched mRNAs in BRF1-IP had consensus AU-rich motifs. Those were not enriched in control IgG-IP. Furthermore, among the top 1500 enriched mRNAs in BRF1-IP, only 33 mRNAs were included in the top 1500 enriched mRNAs in control IgG-IP. We also analyzed Ago2 RIP-RNA-seq data. miR-122 is a liver-abundant miRNA that accounts for 70% of liver total miRNAs. We found that many miR-122 targets, such as *Aldoa*, *Map3k1*, *Ndr3*, *Ccl2* and *Bcl9* were enriched in AGO2-RIP. Again, they were not enriched in control IgG-IP. These results are described on page 13, line 272 to page 14, line 280 of the revised manuscript.

We agree with the reviewer and added a statement about involvement of other RNA binding proteins on page 21, line 437-443 in the revised manuscript. Please also see our response to Reviewer 1 (line 121-123 in this text).

- Figure 6B. In the genome-wide assessment of intron reads, the authors should provide information about how the replicates were treated or the statistical significance of the observed alterations. According to the materials and methods, no normalization by the library size was applied, which could have led to a misinterpretation of the obtained results.

We applied the same normalization method that was used in Figure 3A (mRNA expression, exon reads), to the genome-wide assessment of intron reads. Namely, intron read numbers were normalized by both the total count of reads mapped to the intronic region and the sum of intron lengths (see Materials and Methods). We calculated the means of normalized intron counts from four independent experiments and used them for the scatter and violin plots. We performed a Mann-Whitney U-test and found that there were no mRNAs that show significantly different expression levels between control and *Cnot1-LKO* mice. Therefore, these data were moved to Supplementary Figure 7B and C as references in the revised manuscript. Intron count data were only used to compare distributions of the levels of each GO term between control and *Cnot1-LKO* mice (Supplementary Figure 7C in the revised manuscript). Although there were no significant differences in RNA-seq data, we performed qPCR analysis using intron region-specific primers to examine whether specific pre-mRNAs were differently expressed in livers between control and *Cnot1-LKO* mice (Figure 7B, C in the revised manuscript). These results are described on page 15, line 313 to page 16, line 322 in the revised manuscript.

- Figure 6C. The authors propose that total transcriptional activity is increased in *Cnot1KO* livers. Given the minor difference that is shown and the fact that there are

mRNAs showing opposite behaviours (Figure 6C-G), further evidence should be provided in order to make this statement.

We withdrew the proposal about total transcriptional activity and just reported the results as described in our response to other comments.

Minor comments:

- Figure 2B. Given that the authors claim that most of Cnot1 effects are in immune-related, TF- or apoptosis- mRNAs, it would be an added value to demonstrate that these mRNAs present alterations in their poly(A) length.

We performed poly(A) tail analyses against several mRNAs related to cell cycle, TF, DNA damage, and immune system process using total RNA from livers. The results showed that mRNAs encoding TFs (*Trp53* and *Jun*), DNA damage response-related molecules (*Bbc3* and *Brca1*), and cell cycle regulators (*Cdt1* and *Cdc25a*) had longer poly(A) tails in livers of *Cnot1-LKO* mice (Figure 3D). Poly(A) tail lengths of *Cxcl10* mRNA in livers of *Cnot1-LKO* mice were similar to those in controls, although the band intensity increased, indicating that some immune system process-mRNAs increase regardless of poly(A) elongation. These new experiments are described on page 10, line 188-194 and shown in Figure 3D and Supplementary Figure 2B in the revised manuscript.

- Figure 3C. Is the downregulation in highly expressed metabolic mRNAs specific of metabolic mRNAs? Are highly expressed immune- or apoptosis- mRNAs also downregulated?

Because we noticed that the line that discriminates the two group was not objective, we ceased to divide mRNAs into high and low expression groups. We removed Figure 6C (in the initial manuscript) for the same reason.

- The results section citing figure 5G-I should be revised (text and figures do not match).

We appreciate the reviewer's having pointed this out. We have carefully revised the results. Figures 5 and 6 have been restructured and renumbered in the revised manuscript.

Reviewer 3

1. Fig. 4: More details need to be presented as to how the mRNA half life experiment was done. For example, what time points after actD administration were analyzed? What effect did the actD treatment have on the mice during the course of the experiment? ActD is known, of course, to generate major changes in cellular physiology - thus it is very important for the details of this experiment to be presented so that the data can be put into context.

We agree that our explanation for the mRNA half-life experiment was insufficient. We collected livers at 0, 4, and 8 h after Act. D administration (2 mg/g body weight) to living mice and prepared total RNA. We have added a detailed description in the

Materials and Methods in the revised manuscript. Please see also our response to Reviewer 2 (lines 245-249 and 285-298 in this text). We also added the description, "During 8 h Act. D treatment, we did not observe any obvious abnormalities in mice." in the Materials and Methods in the revised manuscript.

2. The authors note that there is not a strict association between changes in mRNA half-life and changes in mRNA abundance. While they descriptively report on changes in transcription rates, they do not entertain the possibility that this may be due to a 'transcriptional buffering' phenomenon in which the synthetic rate of a transcript is altered by the cell to adjust expression when mRNA decay rates change. If this is indeed the case, it is probably worth at least a paragraph in the discussion to document this buffering phenomenon/mechanism as it is relatively underreported in mammalian systems to date.

According to the reviewer's suggestion, we added brief discussion about the effect of transcriptional buffering on changes in transcription rates of various genes in *Cnot1-LKO* mice on page 20, line 413-416 in the revised manuscript.

Minor Points:

1. Fig 2A/Results section/line 142-143: In the sentence that starts. " In *Cnot1-KO* livers, the population of poly(A) tail length with longer than 70 nt dramatically increased' , I don't really understand what the authors mean by the next phrase, "instead of decrease in that with shorter than 70 nt ." I would recommend rewriting that last part.

We rewrote the sentence on page 8, line 155-157 in the revised manuscript.

2. Fig 4: I'm a bit confused as to what the x axes indicate on the graphs in panels A and C. Please clarify in the legend for the reader.

We have clarified the X axes in the legends of Figures 4A, C, and F.

March 3, 2020

RE: Life Science Alliance Manuscript #LSA-2019-00494-TR

Dr. Tadashi Yamamoto
Okinawa Institute of Science and Technology Graduate University
Cell Signal Unit
1919-1 Tancha
Okinawa 904-0495
Japan

Dear Dr. Yamamoto,

Thank you for submitting your revised manuscript entitled "The CCR4-NOT complex maintains liver homeostasis through mRNA deadenylation". As you will see, the reviewers are now supportive of publication, pending minor revision. We would thus be happy to publish your paper in Life Science Alliance pending the following final revisions:

- Please address the remaining reviewer concerns
- Please make sure that the author order in the manuscript text and our submission system match
- Please link your account in our submission system to your ORCID iD; you should have received an email with instructions on how to do so
- Please enter subject categories in our submission system
- Please upload all figure files, including supplementary figure files, individually
- Please include the supplementary figure legends and supplementary table legends in the main manuscript file
- Please add a callout in the manuscript text to figure 5E
- Please correct the figure legend for figure 6G (currently referred to as "J").

A. FINAL FILES:

B. MANUSCRIPT ORGANIZATION AND FORMATTING:

Sincerely,

Andrea Leibfried, PhD
Executive Editor
Life Science Alliance
Meyerhofstr. 1

69117 Heidelberg, Germany
t +49 6221 8891 502
e a.leibfried@life-science-alliance.org
www.life-science-alliance.org

Reviewer #1 (Comments to the Authors (Required)):

I believe that Takahashi and colleagues have answered the main points raised during the original assessment of the manuscript. In particular, they are more careful with their interpretation of the data.

The manuscript now presents solid data demonstrating that in the absence of Cnot1 in the liver deadenylation is impaired and this affects differentially some families of transcripts. Overall this leads to lethal hepatitis. While the manuscript provides limited mechanistic insights in the molecular function of Cnot1 (that is unlikely to be specific for liver), it reveals a key role of the CCR4-NOT complex in liver maintenance in mice. The revised manuscript may thus be published.

Some minor points will still need to be addressed/corrected before publication. (Those were not present in the original manuscript, and thus that could not be mentioned earlier.)

- Page 7, line 126 and supplementary figure 1E: one would have expected a 1:2 ratio of wild-type and heterozygous mice. Authors should comment on this unanticipated observation.

- Page 13, line 267: "noted" instead of "note"

- Table 1: Format: first comment extends over 2 pages.

Reviewer #2 (Comments to the Authors (Required)):

The authors have improved some limiting technical-aspects of the manuscript and revisited the text, modulating the more mechanistic claims and acknowledging that some of the results could be derived from the immune response to the liver damage. The revised version of the manuscript is, therefore, technically more sound and the conclusions better substantiated by the experiments, if at the expense of limiting the scope of the study.

Still there are some minor changes needed.

- The design and analysis of the genome wide mRNA half-life assessment is now more clear. However, the authors have normalized again to a single mRNA (Rplp0). A genome-wide normalization method (for instance Sedlyarov et al., 2016), would probably be more correct.

The meaning of these sentences is not clear, I would suggest rephrasing them:

- Line 221-222. These data suggest that poly(A) tail elongation and increases in abundance of those mRNAs occurred in hepatocytes of Cnot1-LKO mice.
- Line 380-381. mRNA half-lives were shorter in proportion to decreasing mRNA expression levels in HeLa cells and mouse liver (Maekawa et al, 2015, and this study).
- Line 418-421. When a significant percentage of RNA polymerase II was used to promote transcription of DNA damage response- or immune system process-genes, only a limited percentage of the polymerase may be responsible for transcription of liver function-related genes.

This sentence in the abstract is not fully supported by the data

Line 37-38. The increase of TFs, such as Trp53 and interferon regulatory factor 9 (IRF9), subsequently induces expression of apoptosis-related and inflammation-related mRNAs.

Reviewer 1#

- Page 7, line 126 and supplementary figure 1E: one would have expected a 1:2 ratio of wild-type and heterozygous mice. Authors should comment on this unanticipated observation.

We added sentences describing the observation on page 7, line 127-129.

- Page 13, line 267: "noted" instead of "note"

We corrected the word (page 13, line 267).

- Table 1: Format: first comment extends over 2 pages.

To improve the format problem, we constructed Table 1 in the Word file (and set Table 2 in the same format).

Reviewer #2

The design and analysis of the genome wide mRNA half-life assessment is now more clear. However, the authors have normalized again to a single mRNA (*Rplp0*). A genome-wide normalization method (for instance Sedlyarov et al., 2016), would probably be more correct.

TMM (EdgeR) and RLE (DeSeq2) are general methods for genome wide normalization, but we think that these are not applicable for Act. D chase assay. TMM method normalize count data with a trimmed mean that is the average after removing the upper and lower x% of the data (Robinson and Oshlack, Genome Biol., 2010). RLE method normalize count data with a size factor that is the median of observed counts (Anders and Huber, Genome Biol., 2010). Both normalization methods assume that majority of genes are not differentially expressed. However, the assumption will not be true in Act. D-treated samples, because suppression of transcription results in decrease of many genes. Therefore, normalization with a stable gene would be more appropriate way to analyze our data. We selected *Rplp0* for normalization because mRNAs encoding ribosomal components are stable. Indeed, *Rplp0* mRNA level does not change during 8h Act. D treatment when normalized with 18S rRNA (Supplementary Figure 5 in the re-revised manuscript). These are described on page 12, line 237-241.

Normalization with *Rplp0* mRNA appears to be valid, because there is significant correlation between our half-life data and other three published data as already shown in Supplementary Figure 6A (in the re-revised manuscript). Similar results were obtained

when we used *40S ribosomal protein S3a (Rps3a1)* mRNA for normalization. On the other hand, there was less significant correlation between our data and other three, when we calculated mRNA half-lives following normalization with *Actb*, *Hprt*, or *Gapdh* mRNA levels.

The meaning of these sentences is not clear, I would suggest rephrasing them:

Line 221-222. These data suggest that poly(A) tail elongation and increases in abundance of those mRNAs occurred in hepatocytes of Cnot1-LKO mice.

Line 380-381. mRNA half-lives were shorter in proportion to decreasing mRNA expression levels in HeLa cells and mouse liver (Maekawa et al, 2015, and this study).

Line 418-421. When a significant percentage of RNA polymerase II was used to promote transcription of DNA damage response- or immune system process-genes, only a limited percentage of the polymerase may be responsible for transcription of liver function-related genes.

This sentence in the abstract is not fully supported by the data

Line 37-38. The increase of TFs, such as Trp53 and interferon regulatory factor 9 (IRF9), subsequently induces expression of apoptosis-related and inflammation-related mRNAs.

We rephrased the above four sentences to make the meaning clear and precisely describe the results (lines 218-219, 378-379, 414-417, and 36-39, respectively).

March 14, 2020

RE: Life Science Alliance Manuscript #LSA-2019-00494-TRR

Dr. Tadashi Yamamoto
Okinawa Institute of Science and Technology Graduate University
Cell Signal Unit
1919-1 Tancha
Okinawa 904-0495
Japan

Dear Dr. Yamamoto,

Thank you for submitting your Research Article entitled "The CCR4-NOT complex maintains liver homeostasis through mRNA deadenylation". It is a pleasure to let you know that your manuscript is now accepted for publication in Life Science Alliance. Congratulations on this interesting work.

*****IMPORTANT:** If you will be unreachable at any time, please provide us with the email address of an alternate author. Failure to respond to routine queries may lead to unavoidable delays in publication.*******

DISTRIBUTION OF MATERIALS:

Again, congratulations on a very nice paper. I hope you found the review process to be constructive and are pleased with how the manuscript was handled editorially. We look forward to future exciting submissions from your lab.

Sincerely,
